# Motivational interviewing to support modifiable risk factor change in individuals at increased risk of cardiovascular disease: A systematic review and meta-analysis

Justin Lee Mifsud[1,2]*, Joseph Galea[2], Joanne Garside[3], John Stephenson[4], Felicity Astin[3]

1 Faculty of Health Sciences, University of Malta, Msida, Malta, Europe, 2 Faculty of Medicine and Surgery, University of Malta, Msida, Malta, Europe, 3 Department of Nursing and Midwifery, University of Huddersfield, Huddersfield, United Kingdom, 4 Department of Health Sciences, University of Huddersfield, Huddersfield, United Kingdom

* justin-lee.mifsud@um.edu.mt

**Data Availability Statement:** Data is available in S1 Appendix Table S4.

## Abstract

### Background

Programmes using motivational interviewing show potential in facilitating lifestyle change, however this has not been well established and explored in individuals at risk of, yet without symptomatic pre-existent cardiovascular disease. The objective of this systematic review and meta-analysis was to determine the effectiveness of motivational interviewing in supporting modifiable risk factor change in individuals at an increased risk of cardiovascular disease.

### Methods

Systematic review and meta-analysis with results were reported using the Preferred Reporting Items for Systematic Reviews and Meta-Analyses statement. Health-related databases were searched for randomised controlled trials from 1980 to March 2020. Criteria for inclusion included; preventive programmes, motivational interviewing principles, modification of cardiovascular risk factors in adults of both genders, different ethnicities and employment status, and having at least 1 or more modifiable cardiovascular risk factor/s. Two reviewers independently extracted data and conducted a quality appraisal of eligible studies using an adapted Cochrane framework. The Cochrane framework supports to systematically identify, appraise and synthesize all the empirical evidence that meets the pre-specified eligibility criteria to answer a specific question.

### Findings

A total of 12 studies met the inclusion criteria. While completeness of intervention reporting was found to be adequate, the application of motivational interviewing was found to be insufficiently reported across all studies (mean overall reporting rate; 68%, 26% respectively). No

**Funding:** This study was part of a funded PhD programme by the University of Malta. The funders had no role in study design, data collection and analysis, decision to publish, or preparation of the manuscript.

**Competing interests:** The authors have declared that no competing interests exist.

statistical difference between groups in smoking status and physical activity was reported. A random effects analysis from 4 studies was conducted, this determined a synthesized estimate for standardised mean difference in weight of -2.00kg (95% CI -3.31 to -0.69 kg; p = 0.003), with high statistical heterogeneity. Pooled results from 4 studies determined a mean difference in LDL-c of -0.14mmol/l (5.414mg/dl), which was non-significant. The characteristics of interventions more likely to be effective were identified as: use of a blended approach delivered by a nurse expert in motivational interviewing from an outpatient-clinic. The application of affirmation, compassion and evocation, use of open questions, summarising, listening, supporting and raising ambivalence, combining education and barrier change identification with goal setting are also important intervention characteristics.

## Conclusions

While motivational interviewing may support individuals to modify their cardiovascular risk through lifestyle change, the effectiveness of this approach remains uncertain. The strengths and limitations of motivational interviewing need to be further explored through robust studies.

## Introduction

The European guidelines on cardiovascular disease (CVD) prevention in clinical practice have focused on behaviour change by highlighting and promoting lifestyle therapies, namely; smoking cessation, physical activity as per Joint European Societies' (JES) 5 guidelines [1] and a cardio-protective diet, such as the Mediterranean diet. Adherence to these lifestyle changes is known to reduce CVD risk [2]. Central to these preventive guidelines is the delivery of a person-centred approach. Motivational Interviewing (MI) has been recommended as an intervention to promote lifestyle change in clinical guidelines and is graded as class 1 level A of evidence [2].

The collaborative counselling style contrasts MI to the more directive, expert-driven form of counselling [3]. MI may be adapted to accommodate different culture groups, however the counselling style should hold the core principles and spirit of MI [4]. MI involves reflective listening and understanding the person's views in a non-judgmental, non-biased way without the clinician superimposing their own notions. There are four key principles that form the foundation of MI. First, that the clinician can express empathy. Second that they can promote the client's self-efficacy. Third, that they can recognise resistance or ambivalence expressed by a client about a suggested lifestyle change and 'roll with it rather than wrestle' with it. Fourth, that they can work with their client to help them to notice potential discrepancies between their current circumstances and desired future goals [5, 6]. The principles of MI underpin the development of a therapeutic alliance between the clinician and patient. The 'spirit' of MI is underpinned by partnership, acceptance, compassion and evocation [6] using four overarching processes; engaging, focusing, evoking and planning [6, 7]. The practice of MI involves micro-counselling skills which go by the mnemonic acronym OARS [8]. These include the use of open-ended questions, affirmation, reflective listening, summarizing, informing and advising. By asking open-ended questions, the clinician invites the client to reflect and elaborate further. Affirmation allows the clinician to identify the client's strengths and reflect them back to them to increase their confidence in their own ability to make change (self-efficacy). Reflective listening involves the clinician showing that they fully understand the ideas expressed by a client by reflecting them back to them through paraphrasing the content of the discussion. At the end of

the session, the key points of the discussion are summarized by the clinician in an attempt to provide an overall brief understanding of what has been said. The ability to successfully summarise the key aspects of the discussion also demonstrate active listening and understanding on the part of the clinician. Lastly, the important skill of informing and advising comes into play after having gained the client's consent or if the client asks for further information or advice [7, 8]. Application of these key skills may address ambivalence to change risky behaviour.

Existing studies report MI as an effective intervention used in primary care settings with as few as one MI session of 15–20 minutes reported as being effective in changing behavioural outcomes, including an improvement in modifiable CVD risk factors [9–11]. Moreover, MI has been reported to outperform traditional advice-giving approaches [12]. Consequently, researchers have suggested that clinicians should be trained in using MI skills [11].

There is one systematic review with meta-analysis that provides important information about the effectiveness of MI on primary and secondary prevention of CVD risk factors [13]. The authors concluded that MI could have a favourable effect on efforts to change tobacco smoking habits and improving psychological parameters such as depression and quality of life, compared to usual care. Results for other outcomes were inconclusive and the authors suggested that additional research was required to better understand the optimal format and delivery for MI interventions [13]. Other researchers suggested that primary research should be conducted to determine whether MI can be used with specific groups of individuals 'at increased risk' which could maximise the application and potential impact of this intervention [11]. To date the impact of a MI approach used with individuals at increased risk for CVD, but without established disease, is uncertain as there is limited research on this topic [11, 13]. There does not appear to be a published systematic review that has focused specifically on the effectiveness of MI as an intervention to promote risk factor modification in primary prevention. As previously published systematic reviews [11, 13] have included studies that have recruited both individuals at increased risk of CVD, or diagnosed with CVD. The proposed review specifically focuses on the effectiveness of MI as an intervention to promote risk factor modification in primary prevention and also addresses a gap in the current research by evaluating the characteristics of MI interventions used in clinical trials, including what content is delivered, how and where it is delivered and by whom. In this way the 'active' elements in MI interventions can be considered.

## Review questions

Our review sought to determine the effectiveness of MI intervention in supporting primary prevention through changing modifiable cardiovascular risk factors. Additionally, the review provides an account of the characteristics of MI interventions reported in trials that supported risk factor modification.

The primary and secondary review questions are as follows:

1. Is MI effective in supporting adults at increased risk of cardiovascular disease to make healthy lifestyle changes to reduce cardiovascular risk?

2. What are the characteristics of MI interventions that support risk factor modification?

## Methods

This review is reported using items described in the Preferred Reporting Items for Systematic Reviews and Meta-Analyses (PRISMA) statement [14] (see S1 Table in S1 Appendix). A review protocol can be found in the supplementary information (see S2 Table in S1 Appendix).

## Search strategy

The search strategy was formulated and applied to identify published primary research literature from databases (CINAHL Complete, APA PsycINFO, Academic Search Ultimate, Cochrane Central Register of Controlled Trials, MEDLINE, PubMed,) and electronic journals within health-related resources (E-Journals, Wiley Online Library, PLOS, DynaMed Plus). As Motivational Interviewing was developed in the early 1980s [6], searches were conducted to retrieve peer reviewed articles, published in English, from 1980 to March 2020. Search terms were combined using the Boolean operator OR. Then search terms for each PICO element was combined using the Boolean operator AND. This has ensured that all search terms appear in the record to make the search more focused. Truncations and wildcard symbols were used to broaden the search results. This gave us a comprehensive search strategy to support the identification of relevant studies. For the smaller database (DynaMed) and electronic journals (PLOS) a broad strategy was used, by only using the main search term "motivational interviewing", this was done to ensure completeness of the search. The search strategy is included as supplementary information (see S3 Table in S1 Appendix).

## Study selection

Studies fulfilling the eligibility criteria listed in Table 1 were included. These were studies recruiting adult participants over the age of eighteen, of both genders, representing multiple ethnicities and employment statuses and having at least 1 or more modifiable cardiovascular risk factor/s. The interventions for inclusion consisted of primary prevention interventions, which used MI with the aim to support changes in modifiable cardiovascular risk factors. The comparisons consisted of any other approach used that aimed to support participants to change modifiable cardiovascular risk factors, and did not include MI as part of the intervention. Studies for inclusion were those published between 1980 and March 2020, and limited to randomised controlled trials as reliable sources of evidence [15, 16]. After applying filters (date limiter, peer reviewed, excluding children and adolescents) all article titles and abstracts were screened and duplicates identified and excluded. Studies were assessed for eligibility against the criteria (Table 1). Full text versions of studies meeting the criteria were managed using EndNote software. Reference lists of identified studies were manually searched to identify further potentially eligible publications. Full texts of each eligible study were independently read by two researchers and any disagreements resolved through discussion and where necessary, consultation with a third researcher.

**Table 1. Eligibility criteria as per (PICOs) criteria.**

| Elements | Inclusion | Exclusion |
|---|---|---|
| Population (P) | Adult, aged 18 and over, with at least 1, or more, CVD modifiable risk factor/s | Studies of adults with established CVD |
| Intervention (I) | MI identified as part of a primary preventative intervention programme to enhance modifiable risk factor modification | Studies using any other form of counselling |
| Comparative intervention (C) | Usual care in general practice/other interventions not including MI | Studies in which their comparative intervention includes MI |
| Outcomes (O) | Measurements of modifiable CVD risk factors such as smoking cessation, engagement in physical activities, changes in dietary habits, changes in serum cholesterol and blood pressure status, changes in anthropometric measurements (BMI, weight, waist circumference) | All other form of outcomes and not including measurements of modifiable risk factors such as smoking cessation, engagement in physical activities, changes in dietary habits, changes in serum cholesterol and blood pressure status, changes in anthropometric measurements (BMI, weight, waist circumference) |
| Studies (S) | Randomised controlled studies published between 1980—March 2020 | All other methodological studies |

## Data extraction for study methodology, settings and findings

Data extraction was carried out independently by two researchers using a standardized form which was review specific (see S4 Table in S1 Appendix) [17]. Data was extracted from each study for methodological quality, participant characteristics, total number of participants randomised, setting, country, nature of intervention (MI content, type, frequency, duration), characteristics of the deliverer (professional discipline, training and experience), type of outcomes measured, and relevant findings/results.

## Study outcomes

Effectiveness of the intervention using MI was determined by change in modifiable cardiovascular risk factors (smoking status, dietary eating patterns, physical activity levels, lipid profile levels, blood pressure levels, weight, waist circumference, body mass index).

Data on the characteristics of the MI interventions designed to support CVD risk factor modification were assessed using TIDieR checklist (S5 Table in S1 Appendix) and an MI checklist (S6 Table in S1 Appendix).

## Risk of bias assessment and quality of evidence

Critical appraisal of included studies was undertaken to evaluate the quality of the evidence. An assessment of risk of bias domains was carried out for each individual study [17]. The body of evidence was rated in quality depending on the risk of bias and inconsistency, imprecision, indirectness and publication bias [15, 18]. The GRADE rating was used [15] to determine the fulfilment of key criteria to help to judge the level of confidence that could be placed in the conclusions that were drawn.

## Data synthesis

A statistically significant increase in mean smoking quit attempts, physical activity levels and cardio-protective diet adherence in the intervention group was considered to indicate improvement in CV risk factors. Similarly, a statistically significant decrease in mean blood pressure level, serum cholesterol, weight, waist circumference or body mass index were considered to mark improvements in CV risk factors. Any trends identified across results were also explored. Findings for each outcome were described in a narrative format. Percentage scoring was used for intervention reporting (TIDieR) and reporting of MI elements, as we believe that this would be helpful in synthesizing the overall result of the intervention reporting. To complement the narrative summary the level of heterogeneity across included studies was evaluated to assess the indication for meta-analyses. Should outcomes be sufficiently consistent across studies, unstandardized measures to construct meta-analyses were to be applied. Reflecting the clinical and methodological diversity between the studies a conservative approach to the statistical analysis was planned with a random effect meta-analysis. This was considered as more appropriate than a fixed effects model. The statistical heterogeneity established in the meta analyses is likely to reflect this observed clinical and methodological diversity and suggests that the utilisation of random effect models was appropriate. Stata statistical software (Version 14) was used for the data analysis [19].

## Heterogeneity

Quantitative measures were applied to measure variability between results and determine the level of statistical heterogeneity as measured by values of the $I^2$ statistic in excess of 80%. This is illustrated in forest plots (Figs 2 and 3).

## Results

The systematic search identified a total of 1,968 records. Following the removal of duplicates, 1,668 records remained. A total of 1,592 records were excluded based on a review of the titles and abstracts. Seventy-six full text records were assessed using the parameters of the eligibility criteria. After assessment 64 were excluded. In total 12 studies met the eligibility criteria and were included. The PRISMA flow for study selection and exclusion is illustrated in Fig 1.

### Study characteristics

The randomised controlled trials were conducted in 6 countries: Eight were in Europe: Spain [20], Netherlands [21–24], Denmark [25] and United Kingdom [26, 27]; 2 were in Asia: Taiwan [28] and Malaysia [29]; and 1 in the United States of America [30]. Studies were designed in different ways and MI was used as part of a broader intervention. MI was combined with an individualized healthy lifestyle educative session [28], an educational workbook about hypertension [30], and dietary education and a weight management dietary menu [29]. Other

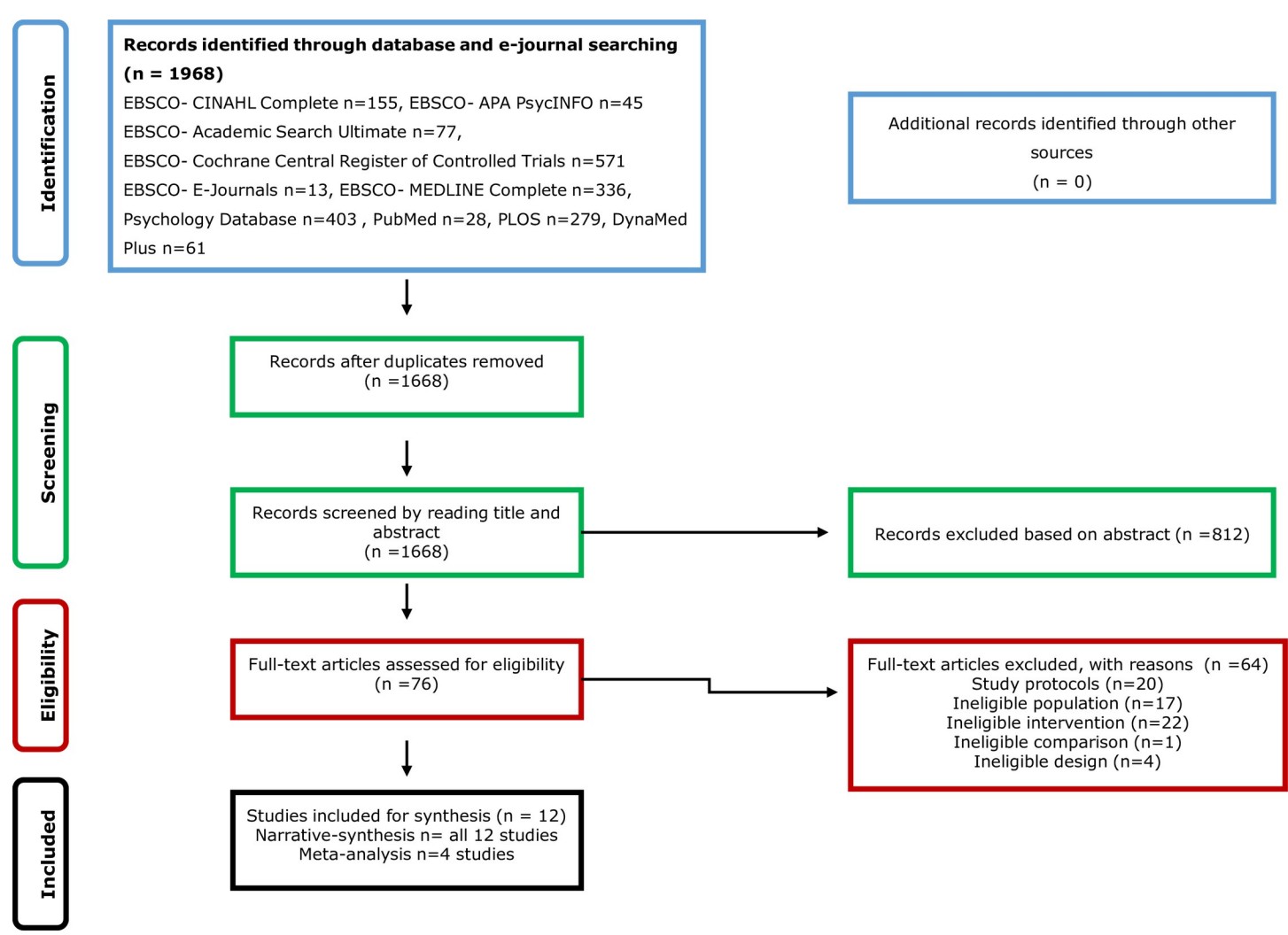

**Fig 1. Prisma flow chart of the study selection and inclusion.**

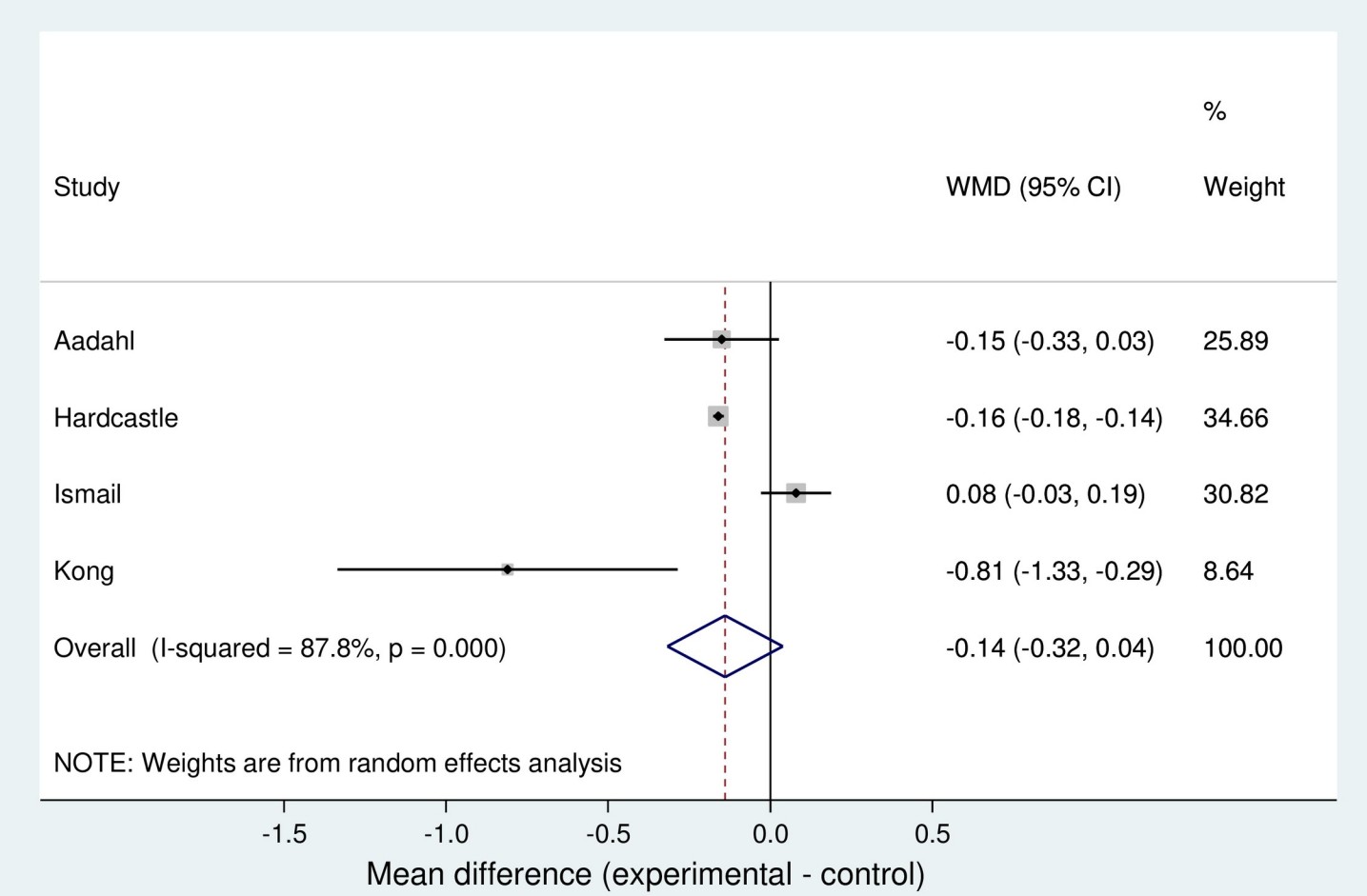

**Fig 2. Forest plot for meta-analysis of LDL-c.**

studies combined MI with an online health risk assessment and tailored feedback [21], risk communication and action planning [31], identification of barriers to change and goal setting [23, 24], behaviour theoretical frameworks [22, 25–27], and clinical dyslipidaemia protocol recommendations [20]. Sample size ranged from 88 [29] to 1742 participants [26]. The number of MI sessions offered, ranged from 1 to 12 sessions and the length of the sessions ranged from 15 to 140 minutes. Seven studies consisted of in-person combined with telephone-based MI [21–24, 28, 30, 31] and 4 studies consisted of face-to-face MI only [20, 25–27, 29]. In 7 studies, sessions took place in community clinics [20, 25–27, 29–31]; other studies used an outpatient clinic [28], an occupational health centre [21], and a diabetes research centre [22]. Two studies did not report the setting [23, 24]. An expert nurse in MI [28], other nurses [22, 25, 31], general practitioners [20], occupational health physicians/nurses [21, 23, 24], licensed dieticians [27, 29] or a physical activity specialist [27], research assistants [30] or health trainers [20] delivered the sessions in all the studies. Training received ranged from 0 to 36 hours of MI training, and only one study had an expert in MI to deliver the session [28]. Five of the RCTs were multicentre trials [20–22, 24, 27, 31]. Ten studies used a 2-group design [20–24, 26, 27, 31] and 2 studies used a 3-group design [24, 25]. A summary of the study characteristics is presented in Table 2.

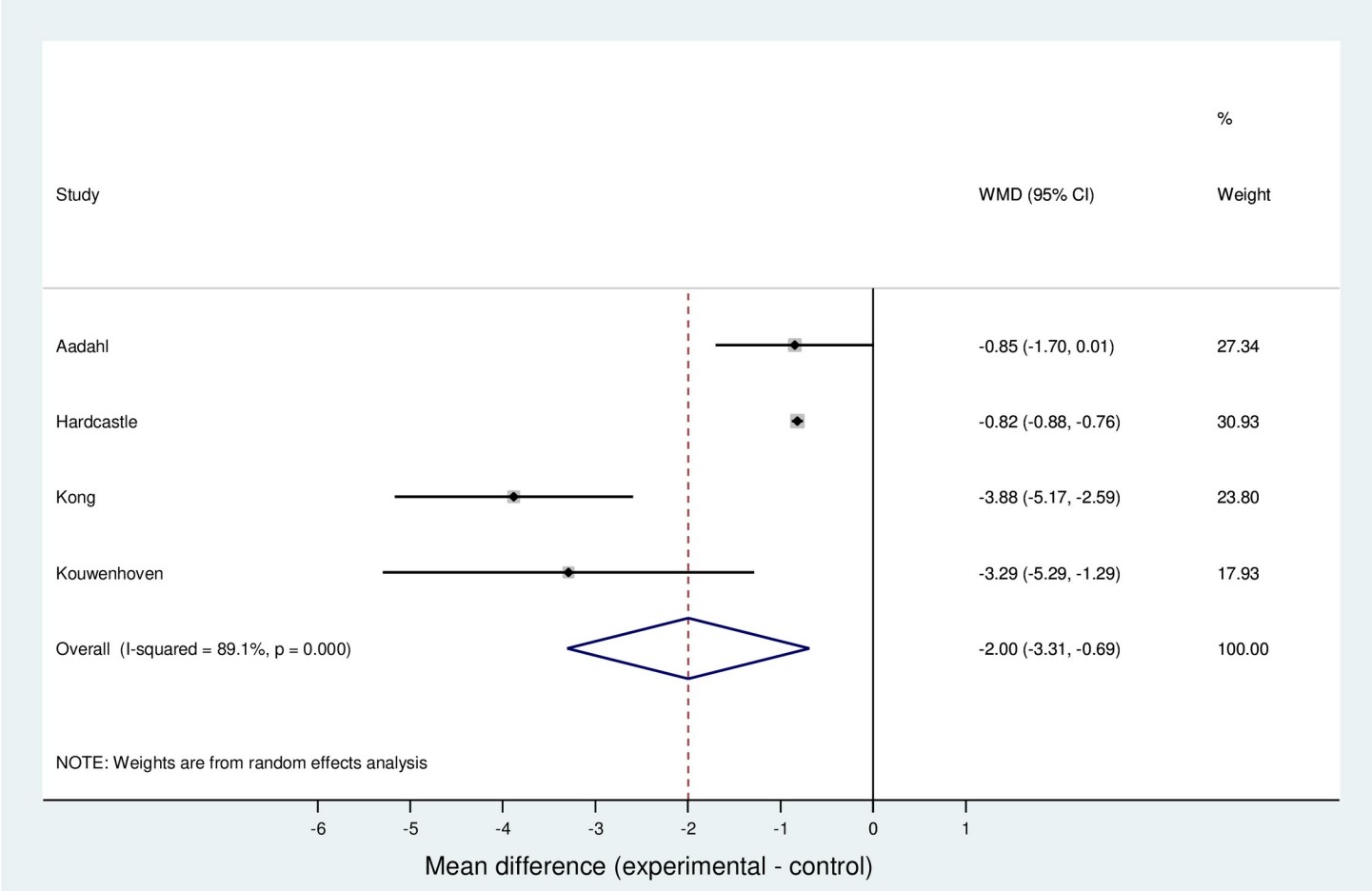

**Fig 3. Forest plot for meta-analysis of weight.**

## Quality appraisal and risk of bias

A computer method to generate the allocation sequence was used by 8 studies [21–28]. Only 6 of the studies prevented risk of selection bias by allocation concealment [22–25, 28, 31]. Lack of blinding of participants and investigators to group allocation was noted in 6 studies [20, 21, 23, 24, 28, 30]. Six studies blinded the assessment of the outcomes to prevent the risk of detection bias [23–28]. Attrition bias was minimalised throughout most studies [20–24, 26–28, 30, 31]. There was no selective reporting in 6 of the 12 studies [20, 23, 24, 26, 28, 31]. The remaining 6, did not provide sufficient detail about the reporting of study outcomes as no protocol was available and a judgement regarding the risk of reporting bias could not be made [22, 25, 27, 29, 30]. For 1 study in particular [21], although a study protocol was made available, it was noted that not all pre-specified outcomes that are of interest in the review were reported in a pre-specified way. Therefore, this study can be indicative of selective reporting, as it allows for reporting bias [21]. Although all authors claimed to use randomization to assign participants to groups, the process may not have been optimal by introducing potential risk of selection bias. The studies have also shown several further weaknesses hindering credibility. For example, the sample size of some studies may have been too small to detect a statistically significant change. Table 3 illustrates a summary of risk of bias across domains.

**Table 2. Characteristics of included studies.**

| Author and year | Country | Participants | Participants randomized | Intervention | Control | Follow-up | Outcomes | Study Design |
|---|---|---|---|---|---|---|---|---|
| Hardcastle (2008), [27] | United Kingdom | Age: 18–65, mean (SD): 51 (1) years. | N = 552 | MI-based approach. | Use of health promotion leaflet. | 6 months | Physical activity, weight, BMI, low- density lipoprotein cholesterol, systolic Bp, diastolic Bp, fruit and vegetable intake | Individual randomisation using blocks, to 1 of the 2 groups |
| | | Gender: 67% females. | | Theory-based (principles & strategies from models of psychotherapy and behaviour change theory). | | | | |
| | | Ethnicity: White. | | Use of open-ended questions and reflective listening. | | | | |
| | | Risk profile: At least with 1 CHD risk factor. | | Different strategies were used depending on an individual's needs and readiness to change. | | | | |
| Koelewijn-van Loon, 2009, [31] | Netherlands | Age: ..., mean (SD): 57 (7) years. | N = 615 | MI-based approach. | Use of risk assessment only & usual nurse led care. | 3 months | Physical activity, smoking cessation, fruit intake. | Multicentre, randomised controlled, using blocks to 1 of the 2 groups |
| | | Gender: 55% females. | | Emphasising reflection on the information received. | | | | |
| | | Ethnicity: White. | | Risk assessment & communication, Use of a Decision support tool (DST). | | | | |
| | | | | Risk profile: one or more CVD risk factors. | | | | |
| Groenewald (2010), [23] | Netherlands | Age:18–65, mean (SD): 46.9 (9.1) years. | N = 816 | MI-based approach. | Use of verbal and written information about their CVD risk profile. | 12 months | Weight, BMI, systolic and diastolic Bp. | Individual randomization, to 1 of the 2 groups |
| | | | | Focus on modification of diet, physical activity and smoking | | | | |
| | | Gender: 100% males. | | Use of open questions, summarizing, listening, supporting, and raising ambivalence. | | | | |
| | | Ethnicity: White. | | | | | | |
| | | Risk profile: CVD 10-year risk score ≥ moderate calculated using Framingham risk score. | | CVD risk communication, action planning using pros and cons of changing the behaviour. | | | | |
| Lakerveld, (2013) [22] | Netherland | Age: mean (SD) 43.6 (5.1) years | N = 622 | MI-based approach | Received existing health brochures. | 12months | Developing T2DM and estimation of CVD risk mortality, self-reported physical activity, fruit and vegetable intake, smoking behaviour. | Multicentre, Randomised, controlled, 2-group |
| | | Gender: 58.4% females. | | Theory-based (theory of planned behaviour). | | | | |
| | | Ethnicity: White Caucasian. | | Problem-solving treatment. | | | | |
| | | Risk profile: with ≥10% estimated risk of T2DM and/or CVD mortality. | | | | | | |
| Aadahl, (2014) [25] | Denmark | Age: 18–69 years; mean (SD) 52.2 (13.8); | N = 166 | MI-based approach; | Instructed to maintain usual lifestyle. | 6 months | Daily sitting. | Single centre, open-ended, controlled, randomised, 2-group. |
| | | Gender: 57% females; | | Theory-based (behavioural choice theory); | | | | |
| | | Ethnicity: White Caucasians; | | Individual behaviour goal-setting, self-efficacy. | | | | |
| | | Risk factor: self-reported 3.5 hours of daily leisure-time sedentary behaviours. | | | | | | |
| Bóveda-Fontán, 2015 [20] | Spain | Age: 40–75 years, mean (SD): 52 (8.59); | N = 227 | MI-based approach; | Consultation delivered by general practitioners who did not receive MI training. | 12 months | Serum cholesterol. | Multicentre, open, controlled, randomised, cluster, 2-group. |
| | | Gender: 62% females; | | Use of a dyslipidaemia protocol. | | | | |
| | | Ethnicity: White Caucasians; | | | | | | |
| | | Risk factor: with dyslipidaemia. | | | | | | |

*(Continued)*

**Table 2.** (Continued)

| Author and year | Country | Participants | Participants randomized | Intervention | Control | Follow-up | Outcomes | Study Design |
|---|---|---|---|---|---|---|---|---|
| Boutin-Foster, (2016) [30] | United States | Age: Mean (SD) 56 (11) years; | N = 238 | MI-based approach; | Received a workbook of strategies on blood pressure control. | 12 months | Blood pressure. | Multicentre, randomised, controlled, in a 1:1 ratio to 1 of the 2 groups. |
| | | Gender: 70% females; | | Positive thinking to enhance core values on a daily basis. | | | | |
| | | Ethnicity: African Americans; | | | | | | |
| | | Risk factor: uncontrolled hypertension. | | | | | | |
| Lin, (2016), [28] | Taiwan | Age: 40+, mean (SD): 63.1 (8.5); | N = 115 | MI-based approach; | Received a single brief lifestyle modification counselling session with a brochure on lifestyle modification; usual care. | 3 months | Physical activity, metabolic syndrome risks. | Single centre, randomised, controlled, with 3-parallel groups. |
| | | Gender: 100% females; | | lifestyle modification program using MI. | | | | |
| | | Ethnicity: White Asian; | | | | | | |
| | | Risk profile: Metabolic syndrome. | | | | | | |
| Kong, (2017), [29] | Malaysia | Age:18–59 years, mean (SD): 34(9) years; | N = 88 | MI-based approach; | Received traditional counselling and weekly aerobic exercise from a medical officer and a Physiotherapist. | 3 months | Weight and waist circumference. | Single-centre, randomised controlled 2 group. |
| | | Gender: 72% females; | | Focus on modification of diet and increase in high intensity interval training. | | | | |
| | | Ethnicity: White Asian; | | | | | | |
| | | Risk factor: BMI of at least 18.5 kg/m2 or above. | | | | | | |
| Kouwenhoven-Pasmooij, 2018 [21] | Netherlands | Age: 40+, mean (SD): 51(6) years; | N = 491 | MI-based approach; | Web-based Health Risk Assessment; | 12 months | Body weight, physical activity, health behaviours, daily intake of vegetables. | Multicentre, randomised, controlled, cluster, 2-group. |
| | | Gender: 15% females; | | Web-based Health Risk Assessment; an additional motivational paragraph in the electronic newsletter; | Personalized suggestions for health promotion; | | | |
| | | Ethnicity: White Caucasian; | | | Electronic newsletter with general information on a healthy lifestyle. | | | |
| | | Risk factor; having at least 1 risk factor (+ve CVD family history, not meeting physical activity target, smoking, self-reported diabetes mellitus or random glucose of ≥ 11.1 mmol/l, obesity, hypertension or the use of antihypertensive drugs; and dyslipidaemia. | | Personalized suggestions for health promotion. | | | | |
| Ismail, (2020), [26] | United Kingdom | Age: 40–74, mean (SD): 69 (4) years; | N = 1742 | MI-based approach; | Use of community-based weight loss, smoking cessation and/or exercise programmes. | 24 months | Physical activity, weight, low-density lipoprotein cholesterol. | Multicentre, randomised controlled, in a 4:3:3 ratio, to 1 of the 3 groups |
| | | | | Theory-based (social cognitive theory, & theory of planned behaviour); | | | | |
| | | | | Focus on modification of diet and physical activity | | | | |
| | | Gender: 14.5% females; | | Use of behaviour change techniques; | | | | |
| | | Ethnicity: White (89.4%); | | workbook, action planning worksheets, case studies, self-monitoring diaries and a pedometer. | | | | |
| | | Risk profile: CVD 10-year risk score ≥20.0% calculated using QRisk2. | | | | | | |
| Groeneveld, 2011, [24] | Netherlands | Age:18–65, mean (SD): 46.9 (9.1) years; | N = 816 | MI-based approach; | Use of verbal and written information. | 12 months | Physical activity, fruit intake. | Individual randomization, to 1 of the 2 groups |
| | | Gender: 100% males; | | Focus on modification of diet, physical activity and smoking | | | | |
| | | Ethnicity: White; | | Use of open questions, summarizing, listening, supporting, and raising ambivalence; | | | | |
| | | Risk profile: CVD 10-year risk score ≥ moderate calculated using Framingham risk score. | | CVD risk communication, action planning using pros and cons of changing the behaviour. | | | | |

Standard deviation (SD), Type 2 diabetes mellitus (T2DM), body mass index (BMI)

## Primary outcome -modifiable cardiovascular risk factor change

Heterogeneity between the reviewed studies made it difficult to pool results and arrive at an overall conclusion. This was due to: substantive differences in how the outcomes were measured across the studies; substantive differences in study parameters outwith reasonable limits of heterogeneity, or unavailable statistical information. As such, the majority of the results had to be interpreted narratively [32]. Where possible, certain parameters, which were not provided, were calculated from others that were given.

**Smoking outcome measurements.** Smoking outcome was measured by 4 studies [20, 22, 24, 31]. Three studies revealed no statistically significant differences between the intervention groups and the control groups [20, 22, 31]. One study found a statistically significant effect at 6 months (OR smoking 0.3, 95%CI 0.1;0.7) but this was not sustained until 12 months follow-up (OR 0.8, 95%CI 0.4; 1.6) [24]. Following MI the number of cigarettes smoked per day reduced significantly across both groups (95% CI: -3.32 to -7.94: mean difference = -5.66: p <0.001), but the difference between groups was non-significant (p = -0.749) [20]. Trend towards smoking cessation in both groups at 6-month and 12-month follow-up was present. However, this change was statistically non-significant [22].

**Dietary outcome measurements.** Dietary outcomes were measured in 6 studies. Mediterranean diet score increased from 8.30 (SD = 2.43) at baseline to 9.41 (SD = 2.47) (MD = 1.11: 95% CI: 1.42 to 7.29: p< 0.001), at 12-month follow-up. However, the difference between intervention and control group was non-significant [20]. In the study by Lakervald,[22], the only group difference was for daily fruit consumption of 0.2 pieces of fruit (95% CI: -0.3 to 0.0, p = 0.05) in favour of the control group, but this was only evident at 6-month follow-up. In the study by Groeneveld, [24] a statistically significant beneficial intervention effect was found for snack and fruit intake, and the effect was sustained at 12 month follow-up. In other studies there was no difference between intervention group and control in dietary changes [21, 27]. On the other hand, between-group significant differences were noted by Kong, Jok [29], in total calorie intake (MD = -553.02, SD = 339.18, CI = -448.64 to -657.41, p = 0.01), dietary fibre intake (MD = 5.11, SD = 0.93, CI = 3.26 to 6.95, p = 0.01), carbohydrate intake (MD =

**Table 3. Risk of bias summary.**

| Authors | Random sequence generation | Allocation concealment | Blinding of participants and personnel | Blinding of outcome assessment | Incomplete outcome data | Selective reporting |
|---|---|---|---|---|---|---|
| | Selection | | Performance | Detection | Attrition | Reporting |
| Hardcastle, 2007 [27] | + | ? | + | + | + | ? |
| Koelewijn-van Loon, 2009 [31] | + | ? | + | ? | + | + |
| Groeneveld, 2010 [23] | + | + | - | + | + | + |
| Groeneveld, 2011[24] | + | + | - | + | + | + |
| Lakervald, 2013 [22] | + | + | + | ? | + | ? |
| Aadahl, 2014 [25] | + | + | + | + | - | ? |
| Boveda-Fonatan, 2015 [20] | ? | - | - | - | + | + |
| Boutin-Foster, 2016 [30] | ? | - | - | - | + | ? |
| Lin, 2016 [28] | + | + | - | + | + | + |
| Kong, 2017 [29] | - | ? | + | ? | ? | ? |
| Kowenhaven-Poamooin, 2018 [21] | + | - | - | - | + | - |
| Ismail, 2020 [20] | + | ? | ? | + | + | + |
| Action +/-/? | + = action performed to reduce risk | | - = action not performed | | ? = insufficient information given | |

-33.23, SD = 10.72, CI = -54.54 to 11.91, p = 0.03), fat intake (MD = -23.29, SD = 4.42, CI = -32.07 to -14.51, p = 0.01) and protein intake (MD = -12.45, SD = 3.41, CI = -19.23 to -5.68, p = 0.365).

**Physical activity outcome measurements.** Physical activity levels were measured in 9 studies. No statistically significant difference between groups were recorded at 3 months [28, 31], at 6 months [24, 25], at 12 months [20–22, 24], and at 24 month follow-up [26]. In the study by Lin [28], when a generalised estimating equation was used, it showed that participants in the MI group had a greater increase in the physical activity levels than the non-MI intervention at 3-month follow-up (MET-min/week = 337, p = 0.02), but no differences were noted when compared to those participants who received the brief intervention [28]. In the study by Hardcastle [27], the MI group were more active, particularly with respect to walking (t = -2.72, P = 0.01). In the study by Bóveda-Fontán [20] and Kouwenhoven-Pasmooij [21] an improvement in both groups was evident, where lack of physical activity was reduced by 96.6% [20], and 50% [21] at the 12-month time point.

**Serum cholesterol outcome measurements.** Serum cholesterol was measured in 5 studies. Significant reductions in total cholesterol levels (MD = -1.3 mmol/l, SD = 0.3, CI = -0.9 to -0.7, p = 0.01), low density lipoprotein cholesterol (MD = -0.8 mmol/l, SD = 0.3, CI = -1.3 to 0.3, p = 0.01) and triglyceride cholesterol (MD = -2.2 mmol/l, SD = 0.2, CI = -2.7 to -1.7, p = 0.01) favoured the motivational intervention group [29]. Significant reductions were also evident in the study by Aadahl [25] for total cholesterol (intervention = -22.7%, control = -1%, p = <0.05) and low density lipoprotein cholesterol (intervention -30.5%, control -11%, p = <0.05). On the other hand, in three studies, participants in the MI group exhibited no significantly greater reduction in total cholesterol, low density lipoprotein cholesterol or triglycerides cholesterol, than the control group at 6 months [27], 12 months [20, 26], and at 24 months [26]. In the study by Boveda, [20], it is interesting to note that when researchers assessed the degree of lipid control by combining those participants who achieved the target total cholesterol and target LDL-c (Tot-c <5.172 mmol/l, LDL-c <3.362 mmol/l) a higher number of patients achieved target figures in the experimental group versus comparator group (13.1% vs 5.0%: adjusted OR = 5.77, 95% CI: 1.67 to 19.91) [20]. Moreover, an overall improvement was observed, with both groups achieving better results in total cholesterol levels (Total sample; MD = -0.51: 95% CI: -0.39 to -0.62 mmol/l: p = 0.001), in low density lipoprotein cholesterol (Total sample MD = -0.36: 95% CI: -0.25 to -0.46 mmol/l: p< 0.001) and triglycerides (Total sample MD = -0.5: 95% CI: -0.3 to -0.7 mmol/l: p< 0.001), but no differences were observed in the high density lipoprotein cholesterol levels (Total sample MD = 0.007: 95% CI: -0.06 to 0.0437 mmol/l: p = 0.309) [20].

**Meta-analysis for LDL-c.** A random effects analysis determined that a synthesized estimate for the unstandardized mean difference in total LDL-c reduction (no intervention vs intervention) was -0.14 mmol/l (95% CI -0.032 to 0.04). A Z-test for overall effect revealed no evidence that the value was non-zero (Z = 1.54, p = 0.124). Individual estimates for the unstandardized mean difference ranged from -0.81 [29] to 0.08 [25]. Cochran's Q test revealed evidence for statistical heterogeneity at the 0.1 significance level (Heterogeneity $x^2_{(3)}$ = 24.5; p< 0.001). The $I^2$ statistic was 87.8%, indicating high statistical heterogeneity. The $T^2$ statistic (extent of between-study variance) was calculated to be 0.0237. The data is summarised in a forest plot showing that overall results favour the intervention in reducing LDL-c (Fig 2).

**Blood pressure outcome measurements.** Four studies measured blood pressure outcomes. Significant group differences favouring the motivational intervention group, in systolic blood pressure (-5.14 mmHg, SD = 2.02, CI = -9.15 to 1.14, p = 0.01) were evident in the study by Kong, [29], and the study by Groenevald, [23] (-0.3 mmHg, CI = -2.8 to 2.2). In the study by Hardcastle, [27], although there was a trend towards improvement, this was nonsignificant.

In contrast, Boutin-Foster [30], found no statistically significant difference in the proportion of participants who had achieved blood pressure control between intervention and control group. Furthermore, the intervention did not prove to be effective in maintaining blood pressure in target range (OR = 1.33, CI: 0.57 to 3.10, p = 0.50), that is <140/90 mmHg, at the 12-month follow-up mark.

**Anthropometric outcome measurements.** Anthropometric outcomes were measured in 8 studies, of which 6 studies exhibited statistical differences between groups [21, 23, 25, 27–29]. Waist circumference decreased amongst participants in the MI group, from 84.2% to 63.2% (p = 0.03) [28]. This resulted in a decrease in the proportion of participants with metabolic syndrome by 18.4% (p = 0.01) at 3 months [28]. The waist circumference of participants also improved in the study by Aadahl, [25] at 6 months, in favour of the MI group (MD = -1.42 cm, 95% CI = -2.54 to -0.29, p = 0.01). This is in line with the study by Kong, [29], where in the MI group (n = 43), waist circumference and body weight decreased by -6.92 cm (SD = 0.87, 95% CI = -8.65 to 5.18, p = 0.01) and -3.35 kg (SD = 0.65, CI = -5.17 to 2.59, p = 0.01) respectively. Percentage reductions for waist circumference and body weight were 8.4% and 6.8% for the MI group (n = 43), versus 1.1% and 0.8% for the control group (n = 45) (p<0.05) [29]. Improved BMI was also evident in the study by Kouwenhoven-Pasmooij, [21], where at 12-month follow-up, there was a statistically significant difference in BMI favouring the intervention group (n = 271); BMI was reduced by 0.69 kg/m$^2$ whilst no reduction was observed in the control group (n = 213).

Conversely, 2 studies found no significant difference between groups in anthropometric outcome measures [20, 26]. However, in the study by Bóveda-Fontán, [20], sub-group analysis showed a significant reduction in the waist circumference and weight of obese and overweight patients from baseline to post intervention (MD = -0.79 cm: 95% CI: -0.287 to -1.746 cm: p = < 0.001; MD = -1.77kg: 95% CI: -0.91 to -2.64 kg p = <0.001) at 12-months [20]. In the intervention group (n = 98), the proportion of obese patients decreased by 8.4% versus 6.7% in the control group (n = 98), indicating a 1.7% difference between groups (McNemar $\chi2$ = 13.899, p = 0.001). Although there was no difference in BMI between the intervention and control groups (p = 0.452), when researchers analysed the total sample (N = 198), it was noted that a BMI difference between groups becomes statistically significant (MD = -0.61 kg/m$^2$: 95% CI: -0.34 to -0.88 kg/m$^2$ p = <0.001) [20].

**Meta-analysis for weight.** In view of the variations in clinical and methodological heterogeneity a random effects analysis was conducted on this outcome. The analysis determined that a synthesized estimate for unstandardized mean difference in total weight reduction (no intervention vs intervention) was -2.00 kg (95% CI -3.31 to -0.69). A Z-test for overall effect revealed strong evidence that the value was non-zero (Z = 2.99, p = 0.003). Individual estimates for the unstandardized mean difference ranged from -0.82 kg [25] to -3.88 kg [29]. Cochran's Q test revealed evidence for statistical heterogeneity at 0.1 significance level (Heterogeneity $x^2_{(3)}$ = 27.4; p< 0.001). The $I^2$ statistic was 89.1%, indicating high statistical heterogeneity, thus implying generalizability. The $T^2$ statistic was calculated to be 1.44. The data is summarised in a forest plot showing that overall results favour the intervention in reducing weight (Fig 3).

**Secondary outcomes- reported intervention elements.** The intervention content reported in the studies was assessed against the template for intervention description and replication (TIDieR) (see S5 Table in S1 Appendix) [33] and an MI checklist (see S6 Table in S1 Appendix). The average of total percentage reporting to at least one of the 12 items across all 12 studies amounted to 68%, highlighting that the overall intervention descriptions were adequately reported (Table 4) and the majority of the included studies may support replicability of the intervention. Reporting for the description of 'what procedures' (item 4), was incomplete in 4 studies [20, 22, 25, 26]. We could not identify any MI elements applied in the intervention arm. In 2 of the studies [23, 24], we could not identify the schedule of intervention delivery

**Table 4. Summary of reported intervention elements.**

| Intervention elements reported in the study | 1. Brief name | 2. Why | 3. What materials | 4. What procedures | 5. Who provided | 6. How | 7. Where | 8. When and how much | 9. Tailoring | 10. Modifications | 11. Planned strategies to maintain fidelity | 12. Extent to which intervention was delivered as planned | 1. Evocation | 2. Developing a change plan | 3. compassion | 4. affirmation | 5. profound acceptance | 6. Open-ended questions | 7. reflection | 8. Rolling with resistance | 9. Eliciting and strengthening change talk | 10. Summarization | 11. Recognizing and reinforcing change talk | 12. Consolidating a client's commitment |
|---|---|---|---|---|---|---|---|---|---|---|---|---|---|---|---|---|---|---|---|---|---|---|---|---|
| Hardcastle, 2007 | - | + | + | + | + | + | + | + | + | + | + | - | - | + | - | + | - | + | + | + | + | - | + | + |
| Koelewijn-van Loon, 2009 | + | + | + | + | + | + | + | + | - | - | - | - | - | + | - | - | + | - | + | - | - | - | - | - |
| Gronevald, 2010 | - | + | + | + | + | + | + | - | - | - | + | - | - | - | - | - | - | + | + | + | - | + | - | + |
| Groeneveld, 2011 | - | + | + | + | + | + | + | - | - | - | + | - | - | - | - | - | - | + | + | + | - | + | - | + |
| Lakervald, 2013 | - | + | + | - | + | + | + | + | - | - | + | - | + | - | - | - | - | - | - | - | - | - | - | - |
| Aadahl, 2014 | - | + | + | + | + | + | + | + | - | - | - | - | - | + | - | - | - | - | - | - | - | - | - | - |
| Boveda-Fonatan, 2015 | - | + | + | + | + | + | + | + | - | - | + | - | - | - | - | - | - | - | - | - | - | - | - | - |
| Boutin-Foster, 2016 | + | + | + | + | + | + | + | + | - | - | - | - | + | + | - | + | - | - | + | - | - | - | - | - |
| Lin, 2016 | - | + | + | + | + | + | + | + | + | - | + | - | + | - | + | + | - | - | - | - | - | - | - | - |
| Kong, 2017 | + | + | + | + | + | + | + | + | + | - | + | - | + | + | - | - | - | - | - | - | + | - | - | - |
| Kowenhaven-Poamooin, 2018 | + | + | + | + | + | + | + | + | + | - | + | + | + | + | + | - | + | + | + | - | - | - | - | - |
| Ismail, 2020 | + | + | + | - | + | + | + | + | - | - | + | + | - | - | - | - | - | - | - | - | - | - | - | - |
| Intervention elements reported, presented as percentages across all 8 studies (%) | 42 | 100 | 100 | 67 | 100 | 100 | 100 | 83 | 33 | 8 | 67 | 16 | 42 | 50 | 16 | 25 | 17 | 33 | 42 | 33 | 8 | 17 | 8 | 25 |
| Mean overall (%) reporting rate | 68% | | | | | | | | | | | | 26% | | | | | | | | | | | |

+ reported,—not reported

(item 8). There was no reporting about tailoring (item 9) in 8 studies [20, 22–26, 30, 31], and modifications (item 10) in 11 studies [20–26, 28–31]. Only 2 of the studies reported testing for fidelity (item 12) [21, 26] (Table 4).

The MI content reported was assessed against a checklist that was developed by the authors and drew upon literature from Miller and Rollnick (see S6 Table in S1 Appendix) [7]. None of the included studies reported all of the 12 expected components of MI [7], and only 1 study used the validated Motivational Interviewing Treatment Integrity code (MITI) [21]. The reported MI components in the studies ranged from 0/12 [20, 26] to 8/12 [27], as shown in Table 4. Developing a change plan appeared to be the most commonly used strategy, evident in 6 studies (50%). Evocation (Drawing out rather than imposing ideas) and reflection appeared to be the second commonly used strategies (42%). These were followed by use of open ended questions (33%), affirmation (25%) and consolidating a client's change talk (25%). Compassion, profound acceptance, rolling with resistance, eliciting, and strengthening change talk, summarization, recognizing and reinforcing change talk, appeared to be rarely evident in the included studies. The average of total percentage reporting to at least one of the 12 MI elements across all 12 studies amounted to only 26% (Table 4).

## Indicators in supporting risk factor modification

Compassion was reported as being used in 2 of the studies; of which programmes showed significant difference effect between groups [21, 28]. Furthermore, evocation which was reported as

being performed in 5 of the reviewed studies, 3 studies showed significant differences in effect between groups [21, 28, 29]. Two studies which used open questions, summarising, listening, supporting and raising ambivalence, also showed significant intervention beneficial effects [23, 24]. Being trained in MI techniques or being an expert, also seemed to be one of the components contributing towards a significant group difference effect. Programmes that reported using MI in conjunction with theoretical frameworks such as the behavioural choice theory or theory of planned behaviour, social cognitive theory and theory of self-regulation appeared to be ineffective [22, 25, 26]. Programmes which used MI combined with education [28], or combined with education and Zumba classes [29], or combined with online health screening with tailored feedback [21], or combined with lifestyle clinical guidelines [20], all had a significant group difference effect. The identified and selected components were categorised according to the study methodological qualities based on our evaluation by using the risk of bias assessment tool [34]. Only those components from moderate to high quality studies are illustrated in Table 5.

## Summary of outcome findings

Findings show that when results were pooled from 4 studies, meta-analyses for LDL-c did not show a statistically significant group difference. From 4 studies, 2 studies exhibited statistically significant group differences in reducing blood pressure [23, 29]. From 8 studies, 5 studies exhibited statistical differences between groups in improving anthropometric outcomes [21, 23, 25, 28, 29]. A meta-analysis from 4 studies demonstrated statistically significant weight reduction favouring the MI intervention group. Findings for the four meta-synthesized outcomes using the GRADE rating [15, 35–38], show that these may not be reliable due to the low quality of evidence (Table 6). The quality level was graded using the GRADE's approach [15].

## Discussion

In our review, group differences in the studies have indicated that programmes using MI as part of their intervention in primary care settings for patients at increased risk of cardiovascular disease may lower serum cholesterol [20, 27, 29], systolic blood pressure [27, 29], metabolic risk [28], and decrease anthropometric measurements [20, 21, 23, 27–29]. These interventions showed significant and clinically effective results within MI intervention groups (participants with dyslipidaemia, having at least 1 risk factor, BMI $\geq$18.5k/m$^2$) in modifying behaviour [20, 21, 23, 29] as well as an equal effect on those with physiological, metabolic and anthropometric conditions [20, 21, 23, 27–29].

Our meta-analysis showed a trend towards LDL-c reduction but this did not reach statistical significance. This is consistent with the work by Lee, [13]. On the contrary to the finding of the study by Lee, [13] our meta-analyses from 4 studies shows evidence, but with limited quality ($\oplus \bigcirc \bigcirc \bigcirc$), for weight reduction favouring the MI intervention group. Our review highlights the notion that application of elements such as compassion, affirmation, evocation, use of open questions, summarising, listening, supporting & raising ambivalence and having the intervention delivered by a nurse expert in MI, or having MI combined with educative resources might yield better results. Barrier change identification and goal setting also seem important elements. Other evidence, however, with quality limitations are: using MI elements with health screening resources and tailored feedback, or having MI applied in conjunction with a set of clinical guidelines. It is also evident that Lin [28] and Groeneveld [23, 24] delivered a programme through a sound study methodology, which consisted of a blended delivery (face to face; telephone). The programme by Groenevald [23, 24] consisted of 3 face to face, and 4 telephone-based sessions lasting between 15 to 60 minutes each. Lin, [28], delivered a one face to face session followed by weekly telephone-based MI calls lasting between 15 to 20

**Table 5. Characteristics of the intervention to support risk factor modification.**

| Intervention characteristics | High quality study (Low risk of bias) showing positive impact/s | Moderate to high quality studies (Low risk of bias) showing no impact/s |
|---|---|---|
| Nature of the program | MI combined with education using a brochure to promote physical activity [28]. | MI programme based on behavioural choice theory [25]. |
| | | MI programme based on theory of planned behaviour and theory of self-regulation [22]. |
| | MI combined with identification of barriers to change & goal setting [23, 24] | MI programme based on social cognitive theory and theory of planned behaviour, using behaviour change techniques, a workbook, action planning worksheets, case studies, self-monitoring diaries and a pedometer [26] |
| Type, frequency, duration, interval | Type- Blended | Type- Blended |
| | Frequency- 13 (1 face to face, 12 telephone-based) | Frequency- 4 (2 face to face and 2 telephone-based) |
| | Time- 15–30 mins each [28] | Time- 30–45 minutes |
| | | Interval- every 6-weeks [25]. |
| | Interval- weekly | Type- Blended |
| | | Frequency- 9 (6 face-to-face, 3 telephone-based) |
| | Type- Blended | Time- 30 min |
| | Frequency- 7 (3 face to face, 4 telephone-based) | Interval- monthly [22]. |
| | | Type- face to face |
| | Time- 15–60 mins each | Frequency- 10 |
| | | Time- 40–120 min |
| | Interval- Not reported [23, 24] | Interval- 1 session/week for the 1st 3 months, followed by 4 sessions delivered at 3, 6, 9 and 12 months [26] |
| MI content | MI consisting of affirmation, compassion, evocation and engagement [28]. | MI consisting of individual behaviour goal settings, self-efficacy enhancement [25]. |
| | MI consisting of open questions, summarising, listening, supporting & raising ambivalence [23, 24] | |
| Characteristics of the deliverer (professional discipline, training and experience) | Professional discipline—Nurse with expertise in MI, Experience- Not reported [28]. | Professional discipline–Nurses, Training and experience–Not reported [25]. |
| | | Professional discipline–Health trainers, |
| | Professional discipline–Occupational physician/nurse | Training and experience–Not reported [26] |
| | Experience- Not reported [23, 24] | |
| Setting | Setting—Outpatient clinic [28]. | Setting—Community clinic [25]. |
| | Setting–Not reported [23, 24] | Setting–Community centres [26] |

minutes each. In the study by Lin, the number of metabolic risks in the MI group was reduced significantly when compared with both brief intervention group and usual care group.

## Study limitations, strengths and generalisability

Although this systematic review attempts to reduce bias by being transparent, rigorous and replicable, there are several limitations at study and outcome level. The first issue is, that this review included English language articles only. Other issues are that the summary of this review is only as reliable as the methods used to test for effectiveness in the included studies.

**Table 6. Programme consisting of MI compared to non-MI programme for individuals at increased risk of CVD.**

| Outcome | MI group vs non-MI group | 95% CI | No of participants | Quality | Comments | Grading the quality across domains |
|---------|--------------------------|--------|--------------------|---------|----------|-------------------------------------|
| Improved LDL-c | Weighted mean difference of -0.14 | CI = -0.32 to 0.04 | N = 2603 (4RCTs) [25–27, 29] | ⊕○○○ Very low | Pooled results favour the intervention but not statistically significant. | Risk of Bias- serious |
| | | | | | | Inconsistency- not serious |
| | | | | | | Indirectness- not serious |
| | | | | | | Imprecision- serious |
| | | | | | | Publication bias- likely |
| Decreased weight | Weighted mean difference of -2.0 | CI = -3.31 to -0.69 | N = 2542 (4RCTs) [21, 25, 29, 39] | ⊕○○○ Very low | Pooled results favour the intervention in reducing weight | Risk of Bias- very serious |
| | | | | | | Inconsistency- not serious |
| | | | | | | Indirectness- not serious |
| | | | | | | Imprecision- not serious |
| | | | | | | Publication bias- likely |

CI: Confidence interval

Quality level and current definitions [15];

High quality ⊕⊕⊕⊕- We are very confident that the true effect lies close to that of the estimate of the effect

Moderate ⊕⊕⊕○ - We are moderately confident in the effect estimate: The true effect is likely to be close to the estimate of the effect, but there is a possibility that it is substantially different.

Low ⊕⊕○○- Our confidence in the effect estimate is limited: The true effect may be substantially different from the estimate of the effect

Very low ⊕○○○- We have very little confidence in the effect estimate: The true effect is likely to be substantially different from the estimate of effect.

Thus, where the quality of the research is possibly contaminated with risk of bias due to inherent problems in the design and its methodology, the results presented in this systematic review need to be interpreted with caution. Heterogeneity between the reviewed studies made it difficult to pool results and arrive at an overall conclusion. This was due to: a wide variation in the context and programme designs as well as differences in how the data outcomes were measured. As such, the majority of the results had to be interpreted narratively [32]. Data such as the application of MI elements was found to be insufficient across the 12 studies and, therefore, it was difficult to detect potential meaningful interactions (mean overall reporting rate 26% to at least one element). Unlike Lee, Choi [13], our review has focused on primary prevention studies only. Our review has not only focused on the effectiveness of MI, but has elaborated on intervention items, such as the characteristics of the intervention delivery. These included the type, frequency, duration, interval from 1 session to the other and setting of sessions, characteristics of the deliverer (professional discipline, training and experience), and the possible mechanisms by which the intervention could have supported risk factor modification. Our review adds to the current MI and lifestyle behavioral change literature, and highlights the likely program intervention components which could work better than other components, acknowledging that, if MI is combined with an educative tool, this might work better. In

addition, an intervention which consists of a blended approach (face-to-face, telephone-based sessions), using short intervals (once weekly call) for 3 months, of about 15–30 minutes each, seems to be the ideal format and dosage of the intervention. Having the intervention delivered by a nurse with expertise in MI, adjusting the focus on affirmation, compassion, evocation, and engagement, are other characteristics and mechanisms by which the intervention could have supported change. As our review has identified these components, there is added value into how new study interventions could be developed and delivered. Our review also highlights the importance of fully reporting comprehensive information about MI intervention components. In addition to Lee, Choi [13], we suggest that if an intervention is not MI compliant; i.e. uses a counselling style approach adapted from MI, then this should be reported. This might encourage researchers to use the available MI compliance assessment tools to establish whether an intervention is MI or a counselling style approach that draws upon some, but not all MI principles and practices [40]. Although, all the included studies evaluated programs using MI to support risk modification in adult individuals at increased CVD risk and of all ethnic origins, application of the evidence must be considered carefully given the methodological heterogeneity of the studies and the outlined review limitations.

## Implications for research

Identifying and understanding the key parameters of interventions is paramount to delivering a preventive program including MI intervention. This systematic review aimed to provide valuable knowledge, which may have useful significance for researchers and clinicians [41, 42]. Firstly, future primary studies should aim to evaluate interventions using standardised measuring tools, with comparable data outcomes. Thus, enabling for pooling of standard results to quantitively synthesize in the case of a systematic review. This will support a conclusive reliable assessment of the intervention effectiveness.

Additionally, the practicality of MI interventions being used in day-to-day clinical practice as well as the cost for its application requires future evaluation. We suggest that MI communication skills (OARS) could be combined with existing resources such as CVD risk calculators. This might act as a triple effect resource: an evaluative, educative and communicative tool, which may further support the modification of cardiovascular risk. MI may be an ideal approach for supporting a specific group of individuals who are at an increased risk of CVD and may likely respond to MI by modifying lifestyle risk factors. Using this approach may be ideal amongst first-degree relatives of CVD patients as they are more likely to have a higher incidence of central obesity, smoking, hypertension and hypercholesterolemia than populations who do not have a biological relative with CVD [43–46]. Therefore, as first-degree relatives of CVD patients generally have multiple risk factors it may be more appropriate to implement MI amongst this group rather than amongst other lower risk populations.

In conclusion, while we adopt a motivational style of counselling for individuals who are at increased risk of developing CVD, the effectiveness of this intervention method remains uncertain as its strengths and limitations require further exploration. As such, programmes using MI may be effective and some intervention components might be more powerful than others in affecting specific cardiovascular risk factor change. Elements such as compassion, affirmation and evocation, if adhered to, could be important mechanisms to establish successful cardiovascular risk factor change in patients at high risk of CVD.

## Supporting information

**S1 Appendix.**
(DOCX)

## Author Contributions

**Conceptualization:** Justin Lee Mifsud, Felicity Astin.

**Data curation:** Justin Lee Mifsud, Joseph Galea.

**Formal analysis:** Justin Lee Mifsud, John Stephenson.

**Funding acquisition:** Justin Lee Mifsud.

**Investigation:** Justin Lee Mifsud.

**Methodology:** Justin Lee Mifsud, Felicity Astin.

**Project administration:** Justin Lee Mifsud.

**Resources:** Justin Lee Mifsud.

**Software:** Justin Lee Mifsud.

**Supervision:** Joseph Galea, Joanne Garside, Felicity Astin.

**Visualization:** Justin Lee Mifsud.

**Writing – original draft:** Justin Lee Mifsud.

**Writing – review & editing:** Justin Lee Mifsud, Joseph Galea, Joanne Garside, John Stephenson, Felicity Astin.

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
