## [Decision Letter · Decision Letter 0]

19 Mar 2020

PONE-D-19-35907

Motivational interviewing to support modifiable risk factor change in individuals at increased risk of cardiovascular disease: a systematic review and meta-analysis.

PLOS ONE

Dear Mr Mifsud

Thank you for submitting your manuscript to PLOS ONE. After careful consideration, we feel that it has merit but does not fully meet PLOS ONE’s publication criteria as it currently stands. Therefore, we invite you to submit a revised version of the manuscript that addresses the points raised during the review process.

Please revise your manuscript. 

As academic editor, my evaluation is that the paper is suitable for publication in* PLOS ONE* subject to changes, as indicated in the reviewers' comments, below. Please address all comments. If you disagree with any comments, please present your arguments/make your case in your rebuttal letter when you submit your revised manuscript. ==============================

We would appreciate receiving your revised manuscript by 19 April 2020. To enhance the reproducibility of your results, we recommend that if applicable you deposit your laboratory protocols in protocols.io, where a protocol can be assigned its own identifier (DOI) such that it can be cited independently in the future. For instructions see: http://journals.plos.org/plosone/s/submission-guidelines#loc-laboratory-protocols

We look forward to receiving your revised manuscript.

Kind regards,

Maggie Lawrence

Academic Editor

PLOS ONE

Journal Requirements:

2. We note that your systematic review search is restricted to articles published until August 2018. Please ensure that that you search is up to date and the systematic review/meta-analysis includes any studies published since then.

5. We note that this manuscript is a systematic review or meta-analysis; our author guidelines therefore require that you use PRISMA guidance to help improve reporting quality of this type of study. Please upload copies of the completed PRISMA checklist as Supporting Information with a file name “PRISMA checklist”.

Reviewers' comments:

Reviewer's Responses to Questions

**Comments to the Author**

1. Is the manuscript technically sound, and do the data support the conclusions?

Reviewer #1: Yes

Reviewer #2: Partly

2. Has the statistical analysis been performed appropriately and rigorously? 

Reviewer #1: I Don't Know

Reviewer #2: No

3. Have the authors made all data underlying the findings in their manuscript fully available?

Reviewer #1: Yes

Reviewer #2: No

4. Is the manuscript presented in an intelligible fashion and written in standard English?

Reviewer #1: Yes

Reviewer #2: No

5. Review Comments to the Author

Reviewer #1: Thank you for the opportunity to review this manuscript (PONE-D-19-35907). The manuscript is a systematic review and meta-analysis to determine the effectiveness of motivational interviewing (MI) for supporting modifiable risk factor change in people at increased risk of cardiovascular disease (CVD).

The abstract is structured using appropriate sub-headings and is concisely written. The initial background section states MI has not been well established and explored (line 56), which is a little misleading because there is a wealth of research investigating the use of MI with various health settings and the paper later refers to a recent systematic review on a similar topic. It might be worth adding some additional context to this statement e.g. what areas are poorly established and have not been explored?

The introduction sets the scene for the review and outlines the role of lifestyle therapies in the prevention of CVD. The introduction also introduces MI and outlines some of the core features of the approach. Understandably this is a fairly concise overview of MI, but I wonder whether it would be worth adding something about how these features are used when working with ambivalence and supporting behaviour change.

The introduction continues by highlighting a gap in the literature and stating how this review compliments other literature. The paper then provides a separate rationale section, which is a little unusual. I wonder whether this rationale content might be better embedded within the introduction section?

The review question is appropriate, but could be a little concise (e.g. Our review sought Line 141). There is also a little blurring between effectiveness, strategies, and mechanisms. I wonder whether it might be clearer to present the aims as three bullet points: 1) effectiveness 2) characteristics 3) mechanisms?

The methods section begins by stating the review aligns with PRISMA, which is good. Although, I could see the attached table in the supporting information section to check completeness. A search strategy section is provided, but this could be more transparent and possibly more accurate. The search strategy is rather difficult to read because of the way the information is presented. The use of brackets to group the databases accessed via EBSCO is useful, but this is not used for databases accessed via ProQuest. I could be wrong, but I also think there is some confusion between databases (e.g. MEDLINE) and database providers (e.g. EBSCO and ProQuest) because there is not a consistent approach to how these are presented. It might be useful to list the databases and put the provider in brackets (e.g. MEDLINE (ProQuest)).

The use of key words/subject headings could also be more explicit and it is unclear whether Boolean operators and/or delimiters were used. I understand an example of the search strategy is provided in the supplementary information, but I was unable to access this. I would also suggest changing the term “concepts” (line 164) to “search terms” or similar. The study selection section refers to RCTs “a high body of evidence”, which is rather unusual language (line 185) and might benefit from re-wording. It also refers to filters being applied when using endnote (line 187), but it is unclear what these filters are and whether they are referring to the delimiters applied during the initial database search.

The data extraction section makes reference to critical appraisal (lines 205-208), but this is rather confusing and might be better re-worded and placed within the risk of bias section further down the paper.

The paper includes a study outcomes sub-heading, which might be unnecessary given the other text provided. This section also does not mention effectiveness, which is one of the main aims of the review.

The results section outlines the search and screening results and provides a PRISMA flowchart. The PRISMA flowchart is a little unusual, but provides the necessary information. The results section then provides a summary of the study characteristics. It would be good to explain the differences in the RCTs (line 265), so readers learn a little more about the studies instead of having to refer to the table. Table two shows the study characteristics, but could be enhanced by providing details of the actual participants (e.g. gender, mean age) instead of reporting the inclusion/exclusion criteria used in the original studies.

The effectiveness of MI in supporting modifiable risk factor change is reported comprehensively. A summary of MI component is also provided, but I am a little unsure how the 12 items were selected and wonder whether a more comprehensive reporting method might be better (e.g. TIDieR DOI: 10.1136/bmj.g1687). It might also be worth considering whether the studies being reviewed used any competence measures (e.g. MITI DOI: 10.1016/j.jsat.2004.11.001).

The discussion and limitation section are clear and make relevant points. The conclusion is brief and accurate. Check wording (line 606) and consider amending “possess”.

Overall, this is a really interesting topic and thorough systematic review. It will be of interest to a wide number of people and help inform practice/research. It does needs some minor revision before publication and could be written more concisely in places.

Reviewer #2: Many thanks for inviting to review this manuscript. Overall,the topic is interesting but there are numerous weaknesses, lack of clarity, poorsynthesis of results and general inconsistencies throughout the manuscript thatmade it difficult to review it to the end. I have picked at sections and willgo ahead to provide some comments. I am happy to re-review the whole manuscriptwhen the authors have attended to the preliminary comments.The manuscript will immensely benefit from professional editingservice prior to resubmission.  

Abstract

Pg 3 line 63-64: It is not clear why only manuscriptsfrom 2013 to August 2018 were explored. What about earlier studies or studiesdone since after August 2018.

Introduction:

The rationale for this review was not effectivelyestablished. Given that a previous review suggested primary research (page 5lines 116 – 118) why implement a review instead of a primary research.

Rationale

Page 6 line 125 – 127 (The impact of MI… has remainedsuboptimal) – provide citation

Search strategy- page 7

It is not clear if the search strategy is adequate. Thereaders will benefit from search strategy/ies implemented in individual database/sand journal/s. EbSCO host a number of databases so a generic strategy cannotapply to all the databases in EBSCO. I wonder what informed the decision to doa separate search in BMJ and Plos Journals and not the numerous psychology andallied health journals. It is also strange that BMJ and Plos journals willreturn lots of articles whereas search in databases returned just a few numbers. It wasnot clear where the conference proceedings were searched (169)

Study Selection –

pg 8Line 183 – what does the authors mean by internationalstudies? Line 185… Not clear what high body of evidence means. Lines 190 – 191… Reference lists of identified studies weremanually search to identify further eligible publication - at what point wasthis done? Was this done only for the 7 included papers?How was disagreement between reviewers regarding studieseligibility dealt with?What was done where it was not declared in a study whetheror not people with established CVD were included?

Dataextraction – pg 9

Lines 205- 207 … Could this be stated as one of thelimitations? Only 3/7 satisfied the items on the risk of bias tables. Again,consideration should be given to the relevance of each of the items to theoverall risk of bias outcomes. E.g. Is it possible to blind a personnel deliveryMI to group allocation?Risk of bias assessment and quality of evidence – pg 10It is great that assessment of risk of bias was done at theoutcome level but were there considerations to the relevance of each domain toindividual outcomes? Could GRADE rating be objectively executed when there notformal meta-analysis in majority of the outcomes? How was inconsistency,imprecision, indirectness and publication bias assessed?

Datasynthesis – pg 10

There are three aspects of heterogeneity – conceptual/clinical,statistical, and methodological. Some aspect of methodological heterogeneity mayhave been dealt with by including only RCT. However, MI in this review werepart of broader interventions in different contexts. Therefore, transparent reporting and justification of how data are grouped for synthesis is essential tocompletely deal with the methodological and conceptual heterogeneity.

Synthesis of results

Apart from the implemented meta-analysis, there were no realattempt at synthesis of results but a mere corroboration of the results fromprimary studies, making this section bulky and cumbersome to understand. Inaddition, implementing alternative metric synthesis means alternative questionswere answered. Although, this was mentioned in passing in the data synthesis section, the authors did not reflect this in reporting result synthesis. Indeed,meta-analysis asks the question of what is the size of the average effect. Onthe other hand, methods like summarising effect estimates only answer thequestion of the range and distribution of the effect. Combining p-values willshow if there is evidence that there is an effect in at least in one study.Lastly, vote counting based on direction(trends) of an effect teases out if thereis evidence of an effect? Therefore, the manuscript will benefit from a real synthesisand reporting that recognises these differences.

Summary of outcome findings

Similar to synthesis of results, the summary of outcomesfindings lumps everything together. For instance, it is not clear what wasbeing compared against MI for each of the outcomes – MI vs other behaviouralchange techniques, or MI vs usual care. In addition, the interpretation of the qualityof evidence did not reflect that each of the synthesis were not from ameta-analysis (as per the diver synthesis methods).

Fig 1 - PRISMA flow diagram

It is strange to see 2724 record excluded only by title reading leaving on 39 for abstract reading. It is typical to do title and abstract screening simultaneously because it may not be clear to remove many studies at the level of the title reading.

6. PLOS authors have the option to publish the peer review history of their article (what does this mean?). If published, this will include your full peer review and any attached files.

Reviewer #1: No

Reviewer #2: Yes: Dr Ukachukwu Abaraogu Physiotherapy and Paramedicine School of Health and Life Sciences Glasgow Caledonian University Scotland United Kingdom

---

## [Author Response · Author response to Decision Letter 0]

19 May 2020

Reviewer’s feedback (reviewer 1)

4. The abstract is structured using appropriate sub-headings and is concisely written. The initial background section states MI has not been well established and explored (line 56), which is a little misleading because there is a wealth of research investigating the use of MI with various health settings and the paper later refers to a recent systematic review on a similar topic. It might be worth adding some additional context to this statement e.g. what areas are poorly established and have not been explored? 

Reply: Thank you for pointing this out. Now I have clarified that this statement is specifically referring to primary prevention. This review, gives focus on individuals without pre-existent cardiovascular disease. 

5. The introduction sets the scene for the review and outlines the role of lifestyle therapies in the prevention of CVD. The introduction also introduces MI and outlines some of the core features of the approach. Understandably this is a fairly concise overview of MI, but I wonder whether it would be worth adding something about how these features are used when working with ambivalence and supporting behaviour change? 

Reply: Thank you for asking this question. I have elaborated on the application of open questions, affirmation, reflective listening, summarizing, informing and advising. This now shows how these features are applied to address ambivalence and support behavior change. 

6. The introduction continues by highlighting a gap in the literature and stating how this review compliments other literature. The paper then provides a separate rationale section, which is a little unusual. I wonder whether this rationale content might be better embedded within the introduction section?

Reply: Thank you for the comment. The rational content is now embedded within the introductory section. 

7. The review question is appropriate, but could be a little concise (e.g. Our review sought Line 141). There is also a little blurring between effectiveness, strategies, and mechanisms. I wonder whether it might be clearer to present the aims as three bullet points: 1) effectiveness 2) characteristics 3) mechanisms? 

Reply: Thank you for this point. Amendments were done as follows:

‘Our review sought to determine the effectiveness of MI intervention in supporting primary prevention through changing modifiable cardiovascular risk factors. Additionally, the review provides an account of the characteristics of MI interventions reported in trials that supported risk factor modification. 

The primary and secondary review questions are as follows:

1) Is MI effective in supporting adults at increased risk of cardiovascular disease to make healthy lifestyle changes to reduce cardiovascular risk?

2) What are the characteristics of MI interventions that support risk factor modification? 

8. The methods section begins by stating the review aligns with PRISMA, which is good. Although, I could see the attached table in the supporting information section to check completeness. A search strategy section is provided, but this could be more transparent and possibly more accurate. The search strategy is rather difficult to read because of the way the information is presented. The use of brackets to group the databases accessed via EBSCO is useful, but this is not used for databases accessed via ProQuest. I could be wrong, but I also think there is some confusion between databases (e.g. MEDLINE) and database providers (e.g. EBSCO and ProQuest) because there is not a consistent approach to how these are presented. It might be useful to list the databases and put the provider in brackets (e.g. MEDLINE (ProQuest)).

The use of key words/subject headings could also be more explicit and it is unclear whether Boolean operators and/or delimiters were used. I understand an example of the search strategy is provided in the supplementary information, but I was unable to access this. I would also suggest changing the term “concepts” (line 164) to “search terms” or similar. 

Reply: The search was retaken. Table showing details (results for each systematic step, use of Boolean operators, truncations and wildcard symbols, application of filters) is now available for each database. The term “concepts” was used when referring to the main term (Coronary heart disease, Motivational interviewing, Adult, Prevention, Risk modification, and Clinical trial). For each concept then there are a number of search terms. For example for Coronary heart disease; "coronary disease*" OR "cerebrovascular disorder*" OR "cardiovascular disease*" OR "cardiovascular disorder*" OR "cerebrovascular disease*" OR "heart disease*" OR "myocardial infarction" OR "heart disease*" OR "coronary*disease" OR "ischemic*disease" OR "athero*" OR "myocardial"….these all fall under this concept. Search strategy is now included in Table S3 of S1 appendix 

9. The study selection section refers to RCTs “a high body of evidence”, which is rather unusual language (line 185) and might benefit from re-wording. 

Reply: We thank the reviewer for pointing this out. The sentence now reads as follows: 

Studies for inclusion were those published between February 2013 and March 2020, but limited to randomised controlled trials as reliable sources of evidence. 

10. The data extraction section makes reference to critical appraisal (lines 205-208), but this is rather confusing and might be better re-worded and placed within the risk of bias section further down the paper. (Page 6) 

Reply: We thank the reviewer for the suggestion. This was now amended and moved under the risk of bias section. 

11. The paper includes a study outcomes sub-heading, which might be unnecessary given the other text provided. This section also does not mention effectiveness, which is one of the main aims of the review.

Reply: We thank the reviewer for the suggestion. We feel that this section now shows the specifics and it was amended as follows 235-250: 

Effectiveness of the intervention using MI was determined by change in modifiable cardiovascular risk factors (smoking status, dietary eating patterns, physical activity levels, lipid profile levels, blood pressure levels, weight, waist circumference, body mass index). 

Data on the characteristics of the MI interventions designed to support CVD risk factor modification were identified; 

1) Reported intervention characteristics: 

• Nature of the intervention 

• Type, frequency, duration, interval

• Characteristics of the deliverer (professional discipline, training and experience)

• Setting

2) Reported MI elements:

• Motivational interviewing content of the intervention

12. The results section then provides a summary of the study characteristics. It would be good to explain the differences in the RCTs (line 265), so readers learn a little more about the studies instead of having to refer to the table. Table two shows the study characteristics, but could be enhanced by providing details of the actual participants (e.g. gender, mean age) instead of reporting the inclusion/exclusion criteria used in the original studies.

Reply: I thank the reviewer for this suggestion. Table 2 was made more explicit. Exclusion criteria was removed and replaced by details such as gender and mean age. More details were included about the RCT design. Line 292-295. 

13. The effectiveness of MI in supporting modifiable risk factor change is reported comprehensively. A summary of MI component is also provided, but I am a little unsure how the 12 items were selected and wonder whether a more comprehensive reporting method might be better (e.g. TIDieR DOI: 10.1136/bmj.g1687). It might also be worth considering whether the studies being reviewed used any competence measures (e.g. MITI DOI: 10.1016/j.jsat.2004.11.001).. 

Reply: We thank the reviewer for this comment. Now the TIDieR checklist is referred to under the subheading ‘Reported intervention elements’ and table 4. line 451-454. 

We do not feel that the TIDieR should replace our previous work, but added to it. Table 5. also reflects to items covered by the TIDieR checklist. These then were categorized according to the study methodological qualities. This will support clinicians/researchers to select intervention components which look promising and likely to succeed. The motivational interviewing content reported was assessed against a checklist that was developed by the authors and drew upon literature from Miller and Rollnick (see Table S6 in S1 appendix). We believe that this highlights the lack of intervention elements reported and the importance of using an MI competence measure. 

14. The discussion and limitation section are clear and make relevant points. The conclusion is brief and accurate. Check wording (line 606) and consider amending “possess”. 

Reply: I thank the reviewer for their comment. This was now changed to ‘adapted’.

Reviewer2: 

15. Pg 3 line 63-64: It is not clear why only manuscripts from 2013 to August 2018 were explored. What about earlier studies or studies done since after August 2018? 

Reply: I thank the reviewer for the comment. Two similar systematic reviews were taken previously. These included searches from January 1999 to December 2009, and from 1980 to February 2013. The difference is that our review has specifically focused on individuals without pre-existent CVD. Now we have updated the search up to March2020. 

16. Introduction: The rationale for this review was not effectively established. Given that a previous review suggested primary research (page 5lines 116 – 118) why implement a review instead of a primary research.

Reply: We thank the reviewer for the comment. The previous review included RCTs up to February 2013, and did not specifically focus on individuals without pre-existent disease. It was decided that firstly a systematic literature review should be conducted. Our review identified 8 RCTs, which were critically appraised using the Cochrane risk of bias tool. Suggestions for a primary study have been put forward throughout this review. A protocol for a primary study is being developed and a primary research study will be tested in the future.

17. Rationale Page 6 line 125 – 127 (The impact of MI… has remained suboptimal) – provide citation

Reply: I thank the reviewer for the comment. Now citations are provided.

18. Search strategy- page 7 It is not clear if the search strategy is adequate. The readers will benefit from search strategy/ies implemented in individual database/s and journal/s. EbSCO host a number of databases so a generic strategy cannot apply to all the databases in EBSCO. I wonder what informed the decision to do a separate search in BMJ and Plos Journals and not the numerous psychology and allied health journals. It is also strange that BMJ and Plos journals will return lots of articles whereas search in databases returned just a few numbers. It was not clear where the conference proceedings were searched (169)

Reply: I thank the reviewer for pointing this out. Search was retaken. Searches were done individually. We have included a Table (see Table S3 in S1 appendix) showing details of the filters applied and the search results. The search strategy section in the manuscript was amended accordingly.

19. Study Selection –pg 8Line 183 – what does the authors mean by international studies? Line 185… Not clear what high body of evidence means. Lines 190 – 191… Reference lists of identified studies were manually search to identify further eligible publication - at what point was this done? Was this done only for the 7 included papers? How was disagreement between reviewers regarding studies eligibility dealt with? What was done where it was not declared in a study whether or not people with established CVD were included?

Reply: We thank the reviewer for the comments. What we meant by ‘international studies’ is, all studies across the globe. However, since this might not have been interpreted clearly, we decided to remove the term ‘international’. 

By having a high body of evidence, we were referring to RCT design. This was amended and now reads line 204-206; 

‘but limited to randomised controlled trials as these are initially considered as the gold standard of evidence (11, 12)’. 

Manual searching of reference lists was done for the 8 identified RCTs.

Any disagreements between the two researchers, were solved through discussion and if necessary, the third researcher was consulted. All studies declared clear inclusion criteria. Only one study was excluded after consulting with the third researcher. This study included participants who underwent coronary angiography. Such a procedure will only be done in symptomatic individuals (positive Exercise tolerance stress test, stable angina, unstable angina, ECG changes). For the latter reason, it was decided to exclude the study and list under ‘secondary prevention’. 

20. Data extraction – pg 9 Lines 205- 207 … Could this be stated as one of the limitations? Only 3/7 satisfied the items on the risk of bias tables. Again, consideration should be given to the relevance of each of the items to the overall risk of bias outcomes. E.g. Is it possible to blind a personnel delivery MI to group allocation? 

Reply: We thank the reviewer for the comments. The following sentence; ‘In our review, critical appraisal for methodological quality was not done to ascertain eligibility status of the studies to be included, but to identify intervention effectiveness of those studies who have used a high-quality methodology’, was now moved under the subheading ‘Risk of bias assessment and quality of evidence’.

We do understand that it is not always possible to blind personal. Such interventions are considered as ‘complex interventions’, hence a double blinded procedure is not always possible. However, we feel that the risk of bias outcome, does reflect the features of the design and conduct of the included studies. 

21. Risk of bias assessment and quality of evidence – pg 10 It is great that assessment of risk of bias was done at the outcome level but were there considerations to the relevance of each domain to individual outcomes? Could GRADE rating be objectively executed when there not formal meta-analysis in majority of the outcomes? How was inconsistency, imprecision, indirectness and publication bias assessed?

Reply: We thank the reviewer for the comments.

We have reconsidered how to go about the individual outcomes. Now the GRADE’s approach was used for meta-analysis (LDL-c, weight) only. We started with a high default rating, since the included studies are RCTs. 

For these 2 outcomes (LDL-c, weight) a decision was made whether to downgrade or upgrade the certainty of the evidence by one or two levels for each factor, leading to a final rating. 

The level of certainty was downgraded by one level for serious concerns about risk of bias or two levels for very serious concerns. The I2 statistic, was used for the meta-analysis, and indicated high statistical heterogeneity for the two outcomes (LDL-c, weight). The results for these outcomes across the studies were consistent and directing towards effectiveness. Therefore, the high statistical heterogeneity could be explained by the variations between the interventions used. This could be seen as a clear reason for heterogeneity. Therefore, it was decided not to downgrade, hence reason why ‘inconsistency’ for ‘Improved LDL-c’ and ‘Decreased weight’ were marked as ‘not serious’. For ‘decreased weight’, as the confidence interval excluded the possibility of no effect, we are confident that an effect is present, so we decided that imprecision was not serious. On the other hand, for ‘Improved LDL-c’, the result was considered imprecise as the confidence interval includes both a meaningful result in one direction and crosses the line of no effect. Also, the confidence interval resulted wide, therefore cannot exclude any meaningful effect in either direction. Hence, the ‘Improved LDL-c’ outcome was downgraded by one level for serious concerns about imprecision. We decided not to downgrade for indirectness, however we made it clear that the results only refer to the population reflected in the systematic review (individuals at an increased risk of CVD).

Publication bias was addressed with the Cochrane RoB (risk of bias) tool. This was considered under the risk of bias/study limitations factor for GRADE.

22. Data synthesis – pg 10

There are three aspects of heterogeneity – conceptual/clinical, statistical, and methodological. Some aspect of methodological heterogeneity may have been dealt with by including only RCT. However, MI in this review were part of broader interventions in different contexts. Therefore, transparent reporting and justification of how data are grouped for synthesis is essential to completely deal with the methodological and conceptual heterogeneity. Synthesis of results- Apart from the implemented meta-analysis, there were no real attempt at synthesis of results but a mere corroboration of the results from primary studies, making this section bulky and cumbersome to understand. In addition, implementing alternative metric synthesis means alternative questions were answered. Although, this was mentioned in passing in the data synthesis section, the authors did not reflect this in reporting result synthesis. Indeed, meta-analysis asks the question of what is the size of the average effect. On the other hand, methods like summarising effect estimates only answer the question of the range and distribution of the effect. Combining p-values will show if there is evidence that there is an effect in at least in one study. Lastly, vote counting based on direction (trends) of an effect teases out if there is evidence of an effect? Therefore, the manuscript will benefit from a real synthesis and reporting that recognises these differences.

Reply: We thank the reviewer for the comments. 

For each outcome (numerical), we planned to extract: 

Number in control group

Mean value of outcome measure in control group

SD of outcome measure in control group

Number in treatment group

Mean value of outcome measure in treatment group

SD of outcome measure in treatment group

A number of outcomes common to 2 or more studies (weight, LDL cholesterol) were observed. Meta analyses were conducted with respect to both of these outcomes. Although the pre-specified outcomes are all measurable outcomes, it was not possible to quantitively synthesize most of the outcomes from the different studies. This is due to either existent and substantive differences in how the outcomes were measured across the studies; substantive differences in study parameters outwit reasonable limits of heterogeneity, or unavailable statistical information. Where possible, certain parameters, which were not provided, were calculated from others that were given: e.g. SD was calculated from mean, confidence interval and number of cases. However, for most of the variables we did not have the means of generating the statistics needed for a particular study for both study groups being compared Therefore we could not include the study in an MA. 

All outcomes that could not be quantitatively synthesized were interpreted narratively. 

We conducted meta analyses using the more conservative random effects models (rather than fixed effects models), reflecting the clinical and methodological diversity between the studies considered, which is identified and acknowledged elsewhere in our paper. The statistical heterogeneity established in the meta analyses is likely to reflect this observed clinical and methodological diversity and suggests that the utilisation of random effect models was appropriate

23. Summary of outcome findings

Similar to synthesis of results, the summary of outcomes findings lumps everything together. For instance, it is not clear what was being compared against MI for each of the outcomes – MI vs other behavioural change techniques, or MI vs usual care. In addition, the interpretation of the quality of evidence did not reflect that each of the synthesis were not from a meta-analysis (as per the diver synthesis methods).

Reply: We thank the reviewer for the comments. We have amended this section. Now references are made available for each of the outcomes. The quality of evidence now reflects the outcomes from the meta-analysis. 

24. Reviewer 1: Overall, this is a really interesting topic and thorough systematic review. It will be of interest to a wide number of people and help inform practice/research. It does need some minor revision before publication and could be written more concisely in places.

25. Reviewer 2: Many thanks for inviting to review this manuscript. Overall, the topic is interesting but there are numerous weaknesses, lack of clarity, poor synthesis of results and general inconsistencies throughout the manuscript that made it difficult to review it to the end. I have picked at sections and will go ahead to provide some comments. I am happy to re-review the whole manuscript when the authors have attended to the preliminary comments. The manuscript will immensely benefit from professional editing service prior to resubmission.

Reply: We thank the reviewers for finding the topic and the results interesting to a wide number of health professionals to support better practice and research in preventive cardiology. 

I would be glad to answer any further questions or comments that you may have.

I look forward to hearing from you,

Thank you for your time,

Justin Lee Mifsud

---

## [Decision Letter · Decision Letter 1]

4 Sep 2020

PONE-D-19-35907R1

Motivational interviewing to support modifiable risk factor change in individuals at increased risk of cardiovascular disease: a systematic review and meta-analysis.

PLOS ONE

Dear Mr Misfud

I am extremely sorry that it has taken me so long to get back to you. Busy workloads (mine and the reviewers’) following COVID 19 restrictions are in part to blame for this. Now that I have had time to consider the review comments of both the original reviewers and the statistical reviewer, I have revisited your paper with these comments in mind. 

I believe this is a good review which has merit, however after revisiting your paper, I find that does not fully meet PLOS ONE’s publication criteria as it currently stands and *major revision is required before it meets publication standards. Some revisions are minor, others are major. I have listed these below. *

We invite you to submit a revised version of the manuscript that addresses the points raised during the review process.

We look forward to receiving your revised manuscript.

Kind regards,

Maggie Lawrence

Academic Editor

PLOS ONE

Additional Editor Comments (if provided):

Dear Mr Misfud

I am extremely sorry that it has taken me so long to get back to you. Busy workloads (mine and the reviewers’) following COVID 19 restrictions are in part to blame for this. Now that I have had time to consider the review comments of both the original reviewers and the statistical reviewer, I have revisited your paper with these comments in mind.

I believe this is a good review and worthy of publication in PLOS One however after revisiting your paper, I find that major revision is required before it meets publication standards. Many are minor, some are more major. I have listed these below.

Abstract

In the Conclusion section that is a result - please remove that.

Main body of the manuscript

line 113 you refer to one systematic review and meta-analysis - please be clear about your meaning here. One review is about primary and secondary prevention, the other is not.

Line 130 you refer to updating previous reviews - this is not accurate - the focus of your current review is narrower, please clarify.

Line 161 after ‘electronic journals’ is everything in brackets an electronic journal or did you look at electronic journals within these resources? Please clarify.

Line 164 I agree with the reviewers, the use of the word ‘concept’ is not useful here. It would be clearer to explain the PICOs elements (to use the term you use in Table 1) and then describe your search in terms of the PICOs elements.

Line 180 are these ‘preventive interventions’ primary prevention interventions? Please clarify.

Line 185 remove the word ‘but’ and replace it with ‘and’.

Line 191 is it possible to conduct searches using Endnote? If not, remove the words ‘retrieved and’.

Line 192 insert the word ‘potentially’ between the 'further’ and ‘eligible’.

Table 1 insert ‘primary’ in your description of the Inclusion criteria for the Intervention element.

line 212 I am aware that the reviewers suggested that you use the TIDieR checklist. The review would be more robust, I think, if you did used TIDieR to report the data extracted. You can add supplementary items, if you felt this was necessary but I haven't identified any items in sections 1) or 2) that would be supplementary. All would fit with TIDieR, item four, in particular. Also, I note you have used TIDieR in the Results section therefore it would make sense to also use it here.

Lines 226 and 228 - swap the order of these two sentences.

Line 223 do you mean ‘data extraction’ rather than ‘data abstraction’? Move the sentence to the data extraction section above and state that it was review specific or ‘adapted from...’

line 250 the heterogeneity paragraph in this location should be about methods only. The results element of this paragraph belongs in the Results section.

Results

Line 256 you report 1114 hits which after duplicates were removed reduced to 860. 792 records were then excluded – leaving a total of 68 - NOT 69 as stated. This mistake is replicated in the PRISMA flow diagram. Please check the numbers and amend both the text and the flow diagram accordingly.

Table 2

Aadahl (2014) record complete the comment in the ‘control’ section

Remove ‘both’ from after ‘gender’ in the Boutin-Foster, (2016) record.

Insert 100% after ‘females’ in the Lin (2016) record

Insert key to acronyms used in the table

Line 303 the meaning here is unclear – either there was or wasn’t selective reporting.

Line 306 surely there was reporting bias in what was reported (Tabel 3 indicates that this is the issue) and, if that is the case, it is not clear why you have singled out one study for comment.

Line 312 replace ‘into’ with ‘to; ‘by’ is not the right word to use in this sentence.

Line 314 – provide an example

Line 322 remove ‘existent and’

Line 324 ‘outwith’ not ‘outwit’

Line 330 here and for all outcomes headings remove the information in brackets from the heading

Line 368 – consider reporting this is such a way as to demonstrate that it is a positive finding.

Line 414 why was a meta-analysis not doe for BP? After a bit of digging, I find it was because the results were reported in different was – it may be worth explaining that here – but also please note, the references are numbered incorrectly. Please check all reference numbering.

Line 459 - what papers are being referred to in this section?

Lines 479 – 481 How do you justify your interpretation of this finding? Is 73% ‘adequate’?

Table 4 The meaning of the second row from the bottom is unclear I.e. Mean overall (%) reporting rate to at least one element

Line 505 I think there is typo - ‘elements’ is used twice, and the meaning is unclear.

Indicators paragraph

This paragraph needs work the English could be improved and the paragraph would perhaps benefit from an explanatory opening sentence. For example, my understanding is that frequency of use of MI elements is mapped against intervention effectiveness. Is that right? This should be described in the methods section and not be making a first appearance here.

Table 5 Presenting this aspect of your findings in this way is a good idea but I would be tempted to reduce this to include only the high-quality study columns - as these will be the results most likely to influence future practice, surely?

The limitations section needs to include the issue of using English language papers only as this is not considered good practice.

Reviewer 2

I note that reviewer 2 continues to argue that since the two previous reviews did not specifically focus on people without pre-existing CVD, findings from them should not represent results specific to people without pre-existing condition. I therefore suggest extending the search to older literature which focused on people without pre-existing CVD. I agree with reviewer 2 and think you either need to follow his suggestion or be clearer about this issue in the body of the text.

Reviewer 3

Please address the issues raised by reviewer 3.

Reviewers' comments:

Reviewer's Responses to Questions

**Comments to the Author**

1. If the authors have adequately addressed your comments raised in a previous round of review and you feel that this manuscript is now acceptable for publication, you may indicate that here to bypass the “Comments to the Author” section, enter your conflict of interest statement in the “Confidential to Editor” section, and submit your "Accept" recommendation.

Reviewer #1: All comments have been addressed

Reviewer #2: (No Response)

Reviewer #3: (No Response)

2. Is the manuscript technically sound, and do the data support the conclusions?

Reviewer #1: Yes

Reviewer #2: Yes

Reviewer #3: Yes

3. Has the statistical analysis been performed appropriately and rigorously? 

Reviewer #1: I Don't Know

Reviewer #2: Yes

Reviewer #3: Yes

4. Have the authors made all data underlying the findings in their manuscript fully available?

Reviewer #1: Yes

Reviewer #2: Yes

Reviewer #3: Yes

5. Is the manuscript presented in an intelligible fashion and written in standard English?

Reviewer #1: Yes

Reviewer #2: Yes

Reviewer #3: Yes

6. Review Comments to the Author

Reviewer #1: (No Response)

Reviewer #2: Authors have done well to address almost all my queries. However, I continue to argue that given the two previous reviews did not specifically focus on people without pre-existing CVD, findings from them should not represent results specific to people without pre-existing condition. I therefore suggest extending the search to older literature which focused on people without pre-existing CVD.

Reviewer #3: interestimg paper.

Abstract

Definition of cochrane framework should be added

Why from 8 studies for pnly 3 was analusis performed?

Authors say that some features may be more likely like ...: how were these variables defined?

Introduction is too long and should be shortened

Methods: did authors performed sub analysis according to grading of paper

Methods: did authors performed sub anakysis for grading of the paper

Methods:authors spoke about conservative effect for random. This should be better specified

Results: plots should be modified and made more consistent

Discussion: high risk patients like hiv may be also interested by this procedure (quote on PMID: 26851703)

7. PLOS authors have the option to publish the peer review history of their article (what does this mean?). If published, this will include your full peer review and any attached files.

Reviewer #1: No

Reviewer #2: **Yes: **Dr Ukachukwu Abaraogu School of Health and Life Sciences Glasgow Caledonian University

Reviewer #3: **Yes: **Fabrizio D'Ascenzo

---

## [Author Response · Author response to Decision Letter 1]

3 Oct 2020

Revision 2. 

Author response

Justin L. Mifsud – Department of Nursing, University of Malta 

Email address: justin-lee.mifsud@um.edu.mt

ORCID iD: https://orcid.org/0000-0001-5380-9418

Re: Motivational interviewing to support modifiable risk factor change in individuals at increased risk of cardiovascular disease: a systematic review and meta-analysis. (Reference number PONE-D-19-35907)

Dear Editor,

I would like to thank you and the reviewers for their helpful comments. I have taken their suggestions on board and did all the possible changes as discussed below. Please find enclosed the edited manuscript in word format. We believe that now the manuscript fully meets PLOS ONE’s publication criteria.

Comments to the Author

1. If the authors have adequately addressed your comments raised in a previous round of review and you feel that this manuscript is now acceptable for publication, you may indicate that here to bypass the “Comments to the Author” section, enter your conflict of interest statement in the “Confidential to Editor” section, and submit your "Accept" recommendation.

Reviewer #1: All comments have been addressed

Reply: Thank you. We have now addressed all the points highlighted by reviewer 2 and reviewer 3.

2. Is the manuscript technically sound, and do the data support the conclusions?

Reviewer #1: Yes

Reviewer #2: Yes

Reviewer #3: Yes

Reply: Thank you

3. Has the statistical analysis been performed appropriately and rigorously?

Reviewer #1: I Don't Know

Reviewer #2: Yes

Reviewer #3: Yes

Reply: Thank you

4. Have the authors made all data underlying the findings in their manuscript fully available?

Reviewer #1: Yes

Reviewer #2: Yes

Reviewer #3: Yes

Reply: Thank you

5. Is the manuscript presented in an intelligible fashion and written in standard English?

Reviewer #1: Yes

Reviewer #2: Yes

Reviewer #3: Yes

Reply: Thank you

Reviewer #2: Authors have done well to address almost all my queries. However, I continue to argue that given the two previous reviews did not specifically focus on people without pre-existing CVD, findings from them should not represent results specific to people without pre-existing condition. I therefore suggest extending the search to older literature which focused on people without pre-existing CVD.

Reply: Thank you, we have now extended the search and included older studies which fit the inclusion and exclusion criteria of our review. 

Reviewer #3: interesting paper.

Reply: Thank you

1. Abstract

Definition of Cochrane framework should be added

Reply: Thank you, we have now included the principles of the Cochrane framework.

2. Why from 8 studies for only 3 was analysis performed?

Reply: Thank you for your comment. In view of variations between studies in measuring the variables, meta-analysis was not always possible. 

3. Authors say that some features may be more likely like ...: how were these variables defined?

Reply: Thank you for your comment. We have deemed these particular intervention characteristics as potential intervention items, after carrying out a careful evaluation of risk of bias and intervention effectives of each study. Studies which had minimal risk and showed significant intervention benefits, their intervention characteristics were identified and selected as more likely to be effective than other characteristics. The selected characteristics will then be used in primary research. 

4. Introduction is too long and should be shortened

Thank you for your comment, we have now deleted some of the word count in the introduction section. 

5. Methods: did authors performed sub analysis according to grading of paper

Reply: By “sub analysis” we assume the reviewers are referring to subgroup analyses as part of the meta analyses. These were not conducted as there were no appropriate strata in the included studies over which results could be presented. We chose to assess statistical heterogeneity with random effects meta analyses. 

6. Methods: authors spoke about conservative effect for random. This should be better specified

Reply: Thank you for your comment. More detail is now included in the manuscript. We conducted meta analyses using the more conservative random effects models (rather than fixed effects models), reflecting the clinical and methodological diversity between the studies considered, which is identified and acknowledged elsewhere in our paper. The statistical heterogeneity established in the meta analyses is likely to reflect this observed clinical and methodological diversity and suggests that the utilisation of random effect models was appropriate.

7. Results: plots should be modified and made more consistent

Reply: Thank you for your comment. We have now modified the plots to include data from the new studies. The plots have been, and remain consistently presented, and they and the accompanying summary text follow a standard format for the presentation of meta analyses results.

Editorial comments

1. Abstract: In the Conclusion section that is a result - please remove that. 

Reply: Thank you for pointing this out. We have edited the conclusion section and moved part of it under the result section. 

2. Main body of the manuscript: line 113 you refer to one systematic review and meta-analysis - please be clear about your meaning here. 

Reply: Thank you for pointing this out. We were referring to the same study which is a systematic review with meta-analysis. Now this has been clarified in the manuscript. 

3. One review is about primary and secondary prevention, the other is not. Line 130 you refer to updating previous reviews - this is not accurate - the focus of your current review is narrower, please clarify.

Reply: Thank you for pointing this out. Now we have made it explicitly clear that our review specifically focuses on primary prevention. Our review, gives focus on individuals without pre-existent cardiovascular disease. For this reason, we agreed to update the search and included older studies which were eligible as per inclusion/exclusion criteria. 

4. Line 161 after ‘electronic journals’ is everything in brackets an electronic journal or did you look at electronic journals within these resources? Please clarify.

Reply: Thank you for pointing this out. Now we have clarified this sentence in the manuscript. This now reads- The search strategy was formulated and applied to identify published primary research literature from databases (CINAHL Complete, APA PsycINFO, Academic Search Ultimate, Cochrane Central Register of Controlled Trials, MEDLINE, PubMed,) and electronic journals within health-related resources (E-Journals, Wiley Online Library, PLOS, DynaMed Plus). 

5. Line 164 I agree with the reviewers, the use of the word ‘concept’ is not useful here. It would be clearer to explain the PICOs elements (to use the term you use in Table 1) and then describe your search in terms of the PICOs elements.

Reply: Thank you for pointing this out. Now we have replaced the term concept by the term ‘search terms’ which links to the PICO elements. 

6. Line 180 are these ‘preventive interventions’ primary prevention interventions? Please clarify.

Reply: Thank you for your comment. We are referring to primary prevention interventions, therefore we now have included the term ‘primary’. 

7. Line 185 remove the word ‘but’ and replace it with ‘and’.

8. Line 191 is it possible to conduct searches using Endnote? If not, remove the words ‘retrieved and’.

9. Line 192 insert the word ‘potentially’ between the 'further’ and ‘eligible’.….. 

Reply: Thank you for the suggestions. We now have amended the manuscript accordingly. 

10. Table 1 insert ‘primary’ in your description of the Inclusion criteria for the Intervention element.

line 212 I am aware that the reviewers suggested that you use the TIDieR checklist. The review would be more robust, I think, if you did used TIDieR to report the data extracted. You can add supplementary items, if you felt this was necessary but I haven't identified any items in sections 1) or 2) that would be supplementary. All would fit with TIDieR, item four, in particular. Also, I note you have used TIDieR in the Results section therefore it would make sense to also use it here.

Reply: We have now referred to the TIDieR under the heading ‘methods’. S1 appendix now shows items checked for each individual study. This is then presented in Table 4. ‘Summary of reported intervention elements’. We used percentage scoring for each item and calculated the overall mean, as we believe that this will make it more interesting for PLOS readers. 

Now we have used TIDieR to report intervention items as shown under the subheading ‘Secondary outcomes- Reported intervention elements. 

Then we moved on to report The MI intervention element. The item checklist to check for MI elements is review specific and draws upon literature from Miller WR, Rollnick S, (Motivational interviewing: Helping people change: Guilford press; 201 2). The results of the checklist used is now available for each study as part of the S1 appendix. Table S5 is the template of the MI checklist. We believe that this checklist highlights the lack of intervention elements reported and the importance of using an MI competence measure. 

11. Lines 226 and 228 - swap the order of these two sentences. Line 223 do you mean ‘data extraction’ rather than ‘data abstraction’? Move the sentence to the data extraction section above and state that it was review specific or ‘adapted from...’

line 250 the heterogeneity paragraph in this location should be about methods only. The results element of this paragraph belongs in the Results section.

Reply: Thank you for the suggestions. We have now amended the manuscript accordingly. 

12. Results: Line 256 you report 1114 hits which after duplicates were removed reduced to 860. 792 records were then excluded – leaving a total of 68 - NOT 69 as stated. This mistake is replicated in the PRISMA flow diagram. Please check the numbers and amend both the text and the flow diagram accordingly.

Reply: Thank you, we have now re-taken the search to extend to older studies. We have changed numerical records accordingly. 

13. Table 2-Aadahl (2014) record complete the comment in the ‘control’ section. Remove ‘both’ from after ‘gender’ in the Boutin-Foster, (2016) record. Insert 100% after ‘females’ in the Lin (2016) record. Insert key to acronyms used in the table

Reply: Thank you for the suggestions. We have now amended the table accordingly. 

14. Line 303 the meaning here is unclear – either there was or wasn’t selective reporting.

Reply: Thank you for pointing this out. We have clarified this statement and it now reads: There was no selective reporting in 6 of the 12 studies

15. Line 306 surely there was reporting bias in what was reported (Tabel 3 indicates that this is the issue) and, if that is the case, it is not clear why you have singled out one study for comment.

Reply: Thank you for your comment. We have commented on 1 particular study, as this study had a protocol available, however it was noted that not all pre-specified outcomes were reported in a pre-specified way. Therefore, this study can be indicative of selective reporting. The remaining 6 studies did not provide sufficient detail about the reporting of study outcomes as no protocol was available. Therefore, judgement with respect to reporting bias could not be carried out. 

16. Line 312 replace ‘into’ with ‘to; ‘by’ is not the right word to use in this sentence. Line 314 – provide an example. Line 322 remove ‘existent and’ Line 324 ‘outwith’ not ‘outwit’ Line 330 here and for all outcomes headings remove the information in brackets from the heading. Line 368 – consider reporting this is such a way as to demonstrate that it is a positive finding.

Reply: Thank you for the suggestions. We have now amended the manuscript accordingly. 

17. Line 414 why was a meta-analysis not doe for BP? After a bit of digging, I find it was because the results were reported in different was – it may be worth explaining that here – but also please note, the references are numbered incorrectly. Please check all reference numbering.

Reply: Thank you for the suggestions. We have now amended the manuscript accordingly. Meta-analysis was not always possible due to variations between studies in the measurements used to calculate same variable/s. Also, we were not able to use the Groeneveld study in the revised meta analysis of weight. This was because although Groeneveld measured weight in both groups at baseline and at 12 months, no stats were provided for the change in weight. Of course, mean weight change can easily be calculated from baseline and follow-up data, but the SD of weight change cannot. We were able to add the other studies and create new plots accordingly. 

18. Line 459 - what papers are being referred to in this section?

Reply: Thank you for pointing this out. We are referring to the studies by Kouwenhaven 2018, Aadahl, 2014 and Kong, 2017. Reference is being made in a forest plot (fig. 3), showing that the synthesized data from the three studies, overall result favour the intervention in reducing weight. 

19. Lines 479 – 481 How do you justify your interpretation of this finding? Is 73% ‘adequate’?

Reply: Thank you for pointing this out. This has now changed since we have included older studies. Now the percentage is 68%. This now reads: 

The average of total percentage reporting to at least one of the 12 items across all 12 studies amounted to 68%, highlighting that the overall intervention descriptions were adequately reported (Table 4) and may, support replicability of the intervention.

20. Table 4 The meaning of the second row from the bottom is unclear I.e. Mean overall (%) reporting rate to at least one element

Reply: Thank you for pointing this out. We have now amended this in the manuscript. This now reads: Mean overall (%) reporting

Line 505 I think there is typo - ‘elements’ is used twice, and the meaning is unclear.

Reply: Thank you, this is now corrected.

21. Indicators paragraph

This paragraph needs work the English could be improved and the paragraph would perhaps benefit from an explanatory opening sentence. For example, my understanding is that frequency of use of MI elements is mapped against intervention effectiveness. Is that right? This should be described in the methods section and not be making a first appearance here.

Reply: Thank you for your comment. Our intension was to identify and select intervention program elements from those studies which showed a significant beneficial between group difference effect. Only elements from high quality studies were selected. The use of these intervention elements was then put forward to be used in primary research. A protocol for a primary study has been drafted and is proposed to be tested.

22. Table 5 Presenting this aspect of your findings in this way is a good idea but I would be tempted to reduce this to include only the high-quality study columns - as these will be the results most likely to influence future practice, surely?

Reply: Thank you for your suggestion. We have now included only the elements from high quality studies. 

23. The limitations section needs to include the issue of using English language papers only as this is not considered good practice.

Reply: Thank you for your comment. This is now included in the limitations section.

Reply: We thank the reviewers for finding the topic and the results interesting to a wide number of health professionals to support better practice and research in preventive cardiology. 

I would be glad to answer any further questions or comments that you may have.

I look forward to hearing from you,

Thank you for your time,

Justin Lee Mifsud

---

## [Editor Report · Decision Letter 2]

12 Oct 2020

Motivational interviewing to support modifiable risk factor change in individuals at increased risk of cardiovascular disease: a systematic review and meta-analysis.

PONE-D-19-35907R2

Dear Mr Mifsud

We’re pleased to inform you that your manuscript has been judged scientifically suitable for publication and will be formally accepted for publication once it meets all outstanding technical requirements.

Kind regards,

Maggie Lawrence

Academic Editor

PLOS ONE

Additional Editor Comments (optional):

Dear Mr Mifsud

It was pleasure to read your paper this afternoon. I apologise again for it being such unextended process - but I believe your efforts and perseverance have resulted in an excellent, interesting paper of relevance to clinicians and researchers. This is robust work with which to underpin your empirical study.
---

## [Editor Report · Acceptance letter]

15 Oct 2020

PONE-D-19-35907R2 

Motivational interviewing to support modifiable risk factor change in individuals at increased risk of cardiovascular disease: a systematic review and meta-analysis. 

Dear Dr. mifsud:

I'm pleased to inform you that your manuscript has been deemed suitable for publication in PLOS ONE. Congratulations! Your manuscript is now with our production department. 

Kind regards, 

on behalf of

Dr. Maggie Lawrence 

Academic Editor

PLOS ONE